# Homeobox A4 suppresses vascular remodeling by repressing YAP/TEAD transcriptional activity

Masahiro Kimura, Takahiro Horie, Osamu Baba, Yuya Ide, Shuhei Tsuji, Randolph Ruiz Rodriguez [ID],
Toshimitsu Watanabe, Tomohiro Yamasaki, Chiharu Otani, Sijia Xu, Yui Miyasaka, Yasuhiro
Nakashima, Takeshi Kimura & Koh Ono[*] [ID]

## Abstract

The Hippo signaling pathway is involved in the pathophysiology of various cardiovascular diseases. Yes-associated protein (YAP) and transcriptional enhancer activator domain (TEAD) transcriptional factors, the main transcriptional complex of the Hippo pathway, were recently identified as modulators of phenotypic switching of vascular smooth muscle cells (VSMCs). However, the intrinsic regulator of YAP/TEAD-mediated gene expressions involved in vascular pathophysiology remains to be elucidated. Here, we identified Homeobox A4 (HOXA4) as a potent repressor of YAP/TEAD transcriptional activity using lentiviral shRNA screen. Mechanistically, HOXA4 interacts with TEADs and attenuates YAP/TEAD-mediated transcription by competing with YAP for TEAD binding. We also clarified that the expression of HOXA4 is relatively abundant in the vasculature, especially in VSMCs. *In vitro* experiments in human VSMCs showed HOXA4 maintains the differentiation state of VSMCs via inhibition of YAP/TEAD-induced phenotypic switching. We generated Hoxa4-deficient mice and confirmed the downregulation of smooth muscle-specific contractile genes and the exacerbation of vascular remodeling after carotid artery ligation *in vivo*. Our results demonstrate that HOXA4 is a repressor of VSMC phenotypic switching by inhibiting YAP/TEAD-mediated transcription.

**Keywords** Hippo signaling; homeobox genes; phenotypic switching; vascular remodeling

**Subject Categories** Development; Signal Transduction; Vascular Biology & Angiogenesis

## Introduction

In spite of extensive effort for the improvement of therapeutic strategies, vascular remodeling is still the major medical problem worldwide, which is closely related to the pathogenesis of atherosclerosis, coronary heart diseases, peripheral artery diseases, and pulmonary hypertension, indicating the need for more detailed understanding of its etiology and underlying molecular mechanisms.

Vascular remodeling is characterized by morphological and structural changes with alterations in cell proliferation, cell migration, and cell death in the vasculature [1], which is mainly composed of vascular smooth muscle cells (VSMCs), endothelial cells, and fibroblasts. Among these cells, VSMCs are located in the midwall of the blood vessels and well-differentiated in the quiescent state with high expression of contractile smooth muscle-specific genes such as SM α-actin (α-SMA), smooth muscle 22-alpha (SM22α), calponin, smoothelin, and SM myosin heavy chain (Myh11) [2]. However, once exposed to inflammatory stimuli or vascular injury, VSMCs transform from a contractile phenotype to a proliferative synthetic phenotype and produce extracellular matrix and matrix metalloproteinases. This phenomenon is called phenotypic switching, which contributes to the formation of vascular remodeling. The regulatory gene programming during this process has been investigated intensively, and recent studies have identified the Hippo signaling effector, Yes-associated protein (YAP) and the transcriptional enhancer activator domain (TEAD) family, as a critical modulator of phenotypic switching [3–5].

The Hippo signaling pathway has been shown to play a central role in controlling organ size by regulating cell proliferation, apoptosis, and stem cell programming [6]. This recently discovered pathway was originally described in *Drosophila* [7] and later found to be conserved in mammals [8]. The Hippo pathway consists of a two-step kinase cascade, in which mammalian sterile 20-like 1/2 (MST1/2) and their scaffold protein salvador homolog 1 (SAV1) phosphorylate large tumor suppressor 1/2 (LATS1/2); then, LATS1/2 and their adaptor protein MOB kinase activator 1A/B (MOB1A/B) phosphorylate YAP in response to various environmental cues such as cell–cell contact [9], mechanotransduction [10], growth factors [11], and metabolic stress [12]. Phosphorylated YAP is inactivated with cytoplasmic retention by 14-3-3 proteins or proteasomal degradation via ubiquitination [13]. On the other hand, unphosphorylated YAP translocates into the nucleus, where it interacts with TEADs to promote downstream gene transcription required for cell growth such as *CTGF* [14], *CYR61* [15], and ankyrin repeat domain 1 (*ANKRD1*) [16]. In this transcriptional complex, YAP lacks a

Department of Cardiovascular Medicine, Graduate School of Medicine, Kyoto University, Kyoto, Japan
*Corresponding author. Tel: +81 75 751 3190; Fax: +81 75 751 3203; E-mail: kohono@kuhp.kyoto-u.ac.jp

DNA-binding domain but works as a transcriptional coactivator, whereas TEADs (TEAD1-4) are transcription factors that bind to the promoter regions of their target genes via a GTIIC sequence (5′-GGAATG-3′) [14,17].

Numerous studies have shown that inappropriate YAP/TEAD transactivation strongly promotes cell proliferation and migration, followed by tissue overgrowth or tumorigenesis [6]. In fact, YAP/TEAD transcriptional activation is a hallmark of many human cancers [18]. Recently, the importance of YAP/TEAD-mediated gene regulation is recognized not only in the cancer field but also in the cardiovascular field [19,20]. Although emerging evidence supports that the YAP/TEAD plays an essential role in VSMC phenotypic switching and subsequent vascular remodeling [3–5], downstream molecules that affect YAP/TEAD transcriptional activity remain to be elucidated.

In the present study, we sought to discover a novel regulator of YAP/TEAD-dependent gene expression. Using an unbiased shRNA screen, we detected Homeobox A4 (HOXA4) as a potent repressor of YAP/TEAD-mediated transcription by competing with YAP for TEAD binding. We also found that expression of HOXA4 is abundant in the vasculature, and HOXA4 maintains the differentiation state of VSMCs via inhibition of YAP/TEAD-induced phenotypic switching, which was confirmed by exacerbated vascular remodeling after carotid artery ligation in *Hoxa4*-deficient mice. This study provides novel molecular insights into vascular remodeling.

## Results

### Pooled shRNA screening identified HOXA4 as a novel negative regulator of YAP/TEAD transcriptional activity

To investigate the transcriptional activity of YAP/TEAD, an 8×GTIIC-luciferase plasmid containing eight TEAD-DNA-binding motifs is commonly used for cell-based reporter assays [10]. Based on this method, we first generated a fluorescent reporter cell line (8×GTIIC-EmGFP-293T; see Materials and Methods) to monitor YAP/TEAD-dependent transcriptional activity (Appendix Fig S1A). This cloned cell line mimicked the transcriptional activity of YAP/TEAD, because EmGFP intensity was faithfully dependent on cell culture density (Fig 1A) or highly responsive to disruption of LATS2, one of the main upstream Hippo pathway components, or forced expression of a constitutively active form of YAP (YAP-S127A) [9] (Fig 1B). We introduced a pooled lentiviral shRNA library consisting of 27,500 shRNAs targeting 5,043 genes (DECI-PHER, Cellecta Inc.) into 8×GTIIC-EmGFP-293T cells at a MOI of 0.3. After cells were kept overconfluent for more than 2 days to maximally repress YAP/TEAD activity at baseline, cells expressing RFP (transduction marker) were sorted into two subpopulations by FACS, "GFP high" or "GFP low", representing 5 or 50% of the cells with highest or lowest GFP signals, respectively. After genomic DNA extraction and PCR amplification, we sequenced shRNAs from the two cell populations using a next-generation sequencer (Fig 1C). Then, we compared the abundance of each shRNA between these two subpopulations and focused on shRNAs targeting negative regulators of YAP/TEAD signaling that were enriched in the "GFP high" sample. After quantile normalization (Appendix Fig S1B and C), the mean frequency of corresponding shRNAs toward each gene was assessed (Fig 1D). We selected 24 candidate genes whose knockdown strongly changed EmGFP intensity as positive hits in the primary screen, and they were validated using the 8×GTIIC-luciferase assay in HEK 293T cells with a different set of shRNAs (Appendix Fig S1D). Because knockdown of HOXA4 upregulated luciferase intensity most significantly (Appendix Fig S1D), we focused further efforts on HOXA4. HOXA4 knockdown in HEK 293T cells not only upregulated luciferase activity but also resulted in the upregulation of *ANKRD1* mRNA, one of the most well-known YAP/TEAD target genes [16] (Fig 1E and F). On the other hand, forced expression of HOXA4 significantly suppressed *ANKRD1* expression (Fig 1G). HOXA4 is a member of the homeobox gene family, which consists of 39 genes assigned to 13 paralog groups on four separate chromosomes. Homeobox genes have spatial and temporal expression patterns during the embryonic period, in which they regulate the development of various organs in corresponding somites as transcriptional factors [21]. Although the functions of homeobox genes in adult have not been fully investigated, some homeobox genes were recently reported to be expressed even in adult organisms and related to various diseases [22,23]. To date, the function of HOXA4 in YAP/TEAD-dependent gene regulation has not been described, but the above data indicated that HOXA4 is a novel repressor of YAP/TEAD transcriptional activity.

### HOXA4 negatively regulates YAP/TEAD transcriptional activity independent of YAP phosphorylation

HOXA4 knockdown also resulted in the upregulation of other known YAP target genes, such as *CTGF*, *CYR61*, and *BIRC5* (Fig 2A and Appendix Fig S2A) [24]. On the other hand, forced expression of HOXA4 reduced the transcription of these genes except for *BIRC5* (Appendix Fig S2B and C). We next determined whether HOXA4 acts through the canonical Hippo kinases to regulate YAP. Because the expression of Hippo signaling kinases was unaffected by HOXA4 knockdown, these genes are not direct transcriptional targets of HOXA4 (Appendix Fig S3). HOXA4 reduced 8×GTIIC-luciferase reporter responses even in LATS1/2 knockdown cells; therefore, HOXA4 seemed to work in the downstream of the Hippo signaling cascade (Fig 2B). YAP with mutations in all five LATS kinase phosphorylation motifs (YAP-5SA) constitutively stayed in the nucleus and promoted the transcription of downstream genes [9]. Unexpectedly, forced HOXA4 expression significantly suppressed YAP-5SA-induced gene expression (Fig 2C). Moreover, knockdown of endogenous HOXA4 further promoted YAP-5SA-induced gene expression (Fig 2D). Therefore, the suppression of YAP/TEAD transcriptional activity by HOXA4 seemed to be entirely independent of YAP phosphorylation. We also confirmed that overexpression of HOXA4 significantly suppressed YAP target genes such as *CTGF* and *CYR61* (Fig EV1A) and that shHOXA4 significantly upregulated these genes (Fig EV1B). TAZ is a paralog of YAP [25]. TAZ target genes such as *CTGF* and *CYR61* were enhanced by shHOXA4 and suppressed by HOXA4 (Fig EV1C).

### HOXA4 physically interacts with TEAD but not YAP

Several genes have been reported to inhibit YAP/TEAD activity via protein–protein interactions such as VGLL4-TEADs [26], HNF

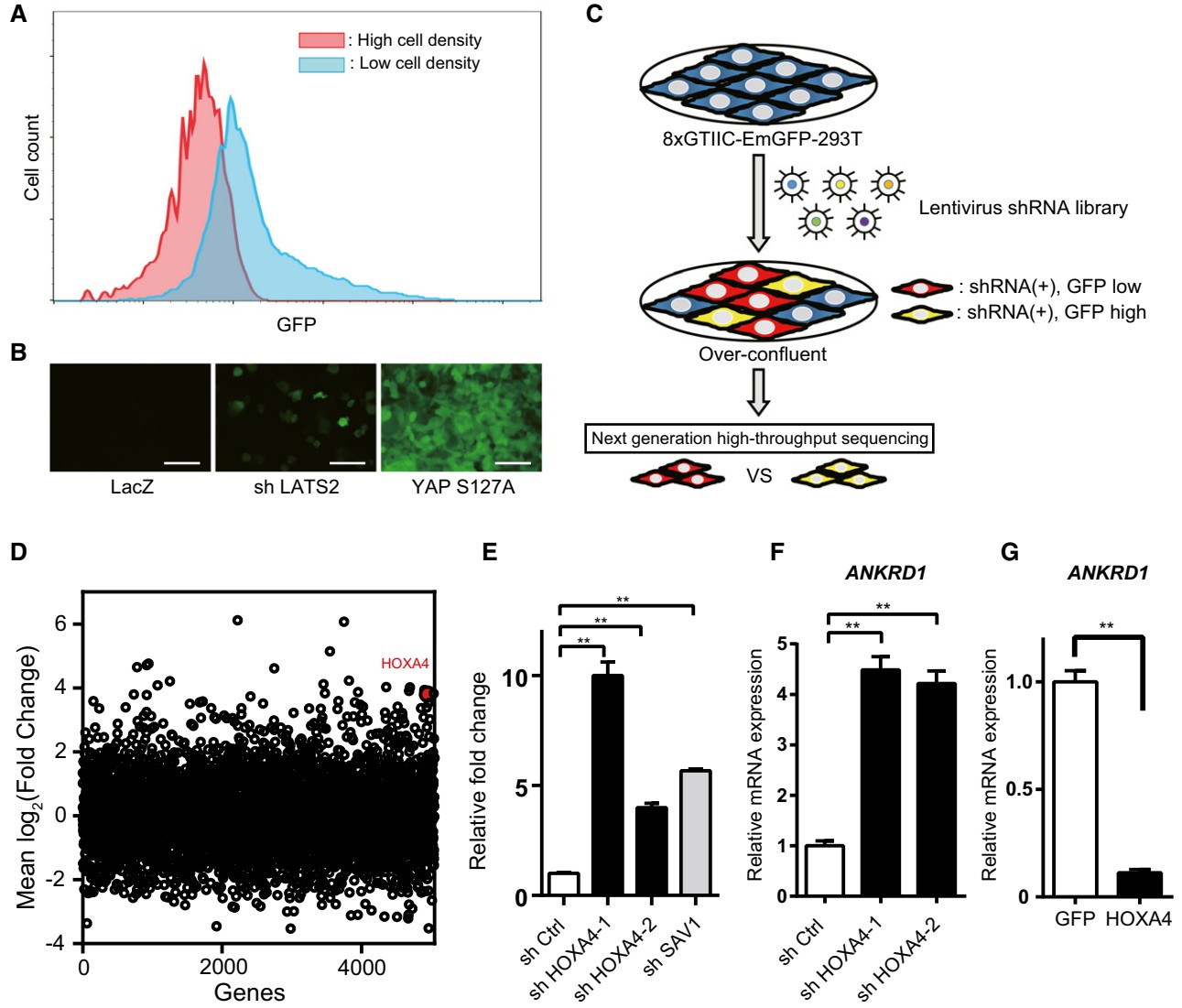

**Figure 1. Identification of HOXA4 as a negative regulator of YAP/TEAD transcriptional activity through a pooled shRNA screen.**

A Dynamic fluorescence patterns according to cell density in 8×GTIIC-EmGFP 293T cells.

B EmGFP intensity in LATS2 knockdown or forced expression of a constitutively active form of YAP (YAP-S127A). Scale bars indicate 100 μm.

C Schematic drawing of FACS-based pooled shRNA screen of molecules that affect the YAP/TEAD signaling pathway in 8×GTIIC-EmGFP 293T cells.

D Scatter plots of mean frequency of each shRNA per gene. HOXA4 is indicated as a red dot.

E Augmented 8×GTIIC-luciferase reporter activity by loss of HOXA4. ShSAV1 was used as an experimental control. **$P < 0.01$ versus shCtrl, by ANOVA with Sidak's correction.

F, G Quantitative real-time PCR analysis of *ANKRD1* expression in HEK 293T cells transfected with shHOXA4 (F) or HOXA4 expression vector (G). **$P < 0.01$, by unpaired two-tailed Student's *t*-test.

Data information: Graphs of (E–G) show mean ± SEM; three biological repeats.

4α-TEAD4 [27], and RUNX3-TEAD4-YAP complexes [28]. Because HOXA4 persistently stays in the nucleus without affecting YAP subcellular localization (Figs EV1D and 1E), we speculated that HOXA4 might physically interact with TEADs or YAP. To examine this possibility, we performed several protein–protein interaction experiments. In HEK 293T cells co-transfected with tagged HOXA4 and TEAD1 or YAP, the co-immunoprecipitation (co-IP) results indicated that HOXA4 interacts with TEAD1 (Figs 3A and EV2A), but not YAP (Fig 3B). HOXA4 also interacted with other three isoforms of TEAD (Figs EV2B and 2C). We also investigated the potential interaction between endogenous proteins in HEK 293T cells and endogenous immunoprecipitation of TEAD1 pulled down HOXA4 and YAP (Fig 3C). To map the HOXA4-TEAD1 interaction domains, several truncated mutants of HOXA4 and TEAD1 were generated. Sequential co-IP assays revealed HOXA4 binds to TEAD1 via HOXA4 amino acids (a.a.) 216–274, which is homeodomain (Fig 3D), and

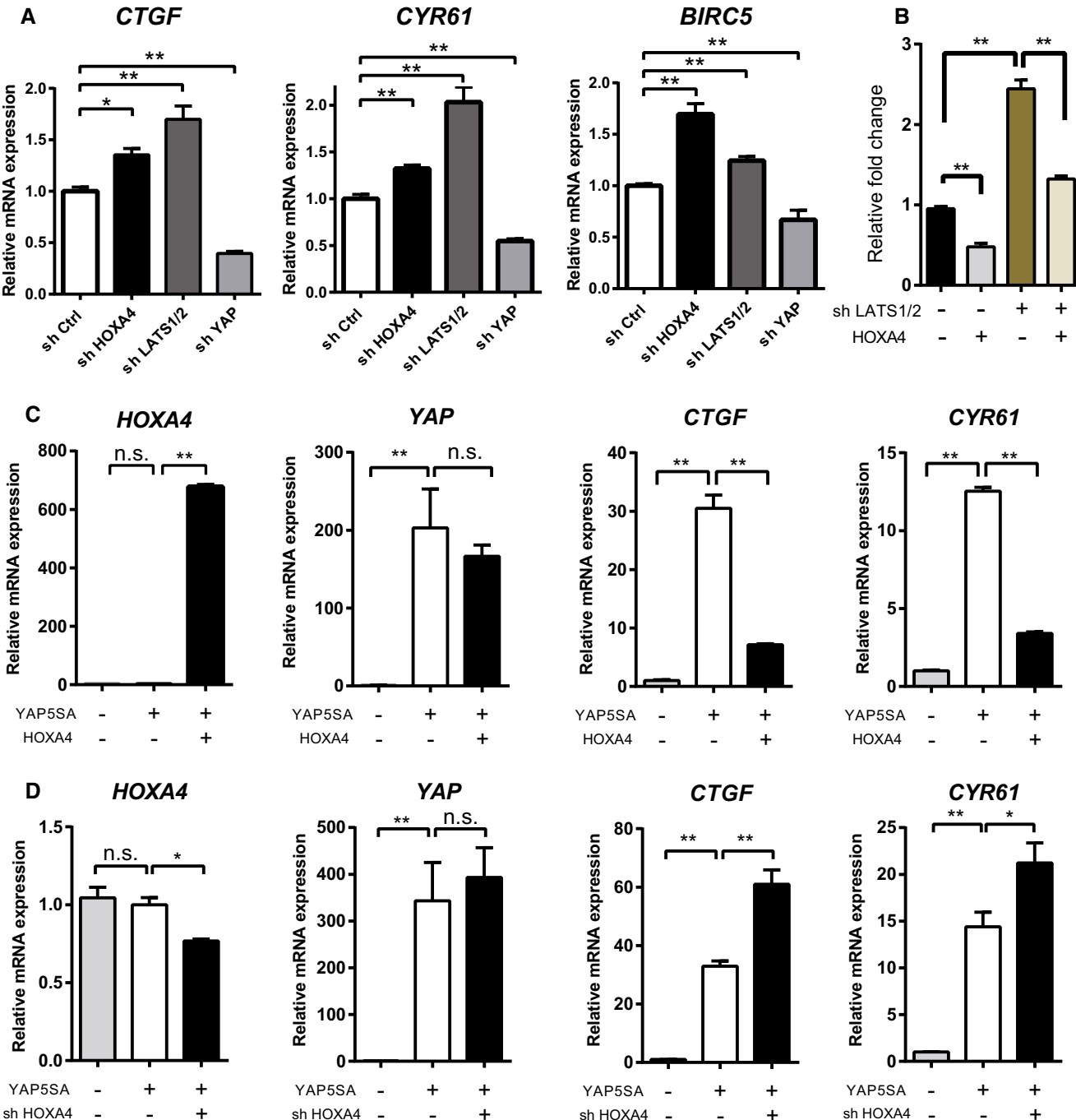

**Figure 2. HOXA4 negatively regulates YAP/TEAD transcriptional activity independently of YAP phosphorylation.**

A  Quantitative real-time PCR analysis of *CTGF*, *CYR61*, and *BIRC5* in HEK 293T cells with knockdown of endogenous HOXA4. ShLATS1/2 and shYAP were used as an experimental control. *$P < 0.05$, **$P < 0.01$ versus shCtrl, by ANOVA with Sidak's correction.

B  8×GTIIC-luciferase reporter assays in HEK 293T cells transfected HOXA4 and/or shLATS1/2. **$P < 0.01$ versus shCtrl, by two-way ANOVA with Sidak's correction.

C  Quantitative real-time PCR analysis of *HOXA4*, *YAP*, *CTGF*, and *CYR61* in HEK 293T cells with overexpression of HOXA4 and/or constitutive active form of YAP (YAP -5SA). *$P < 0.05$, **$P < 0.01$ versus YAP-5SA with LacZ, by ANOVA with Sidak's correction.

D  Quantitative real-time PCR analysis of *HOXA4*, *YAP*, *CTGF*, and *CYR61* in HEK 293T cells with knockdown of endogenous HOXA4 and/or overexpression of YAP-5SA. *$P < 0.05$, **$P < 0.01$ versus YAP-5SA with shCtrl, by ANOVA with Sidak's correction.

Data information: All data are presented as mean ± SEM; three biological repeats.

the N-terminal region of TEAD1 (Fig 3E). Finally, TEAD1 a.a. 30–135 was successfully co-immunoprecipitated by HOXA4 a.a. 216–274 (Fig 3F), whereas TEAD1 a.a. 30–102, only TEA domain, failed to do so (Fig 3G). Taken together, these results indicated that HOXA4 inhibits YAP/TEAD transcriptional activity via its interaction with TEADs (Fig 3H and I).

**HOXA4 competes with YAP for binding to TEADs to suppress the expression of YAP/TEAD target genes**

The interaction feature of HOXA4-TEADs implies two possible mechanisms for the inhibition of YAP/TEAD transcriptional activity. One is HOXA4 might compete with YAP for binding to TEADs, and

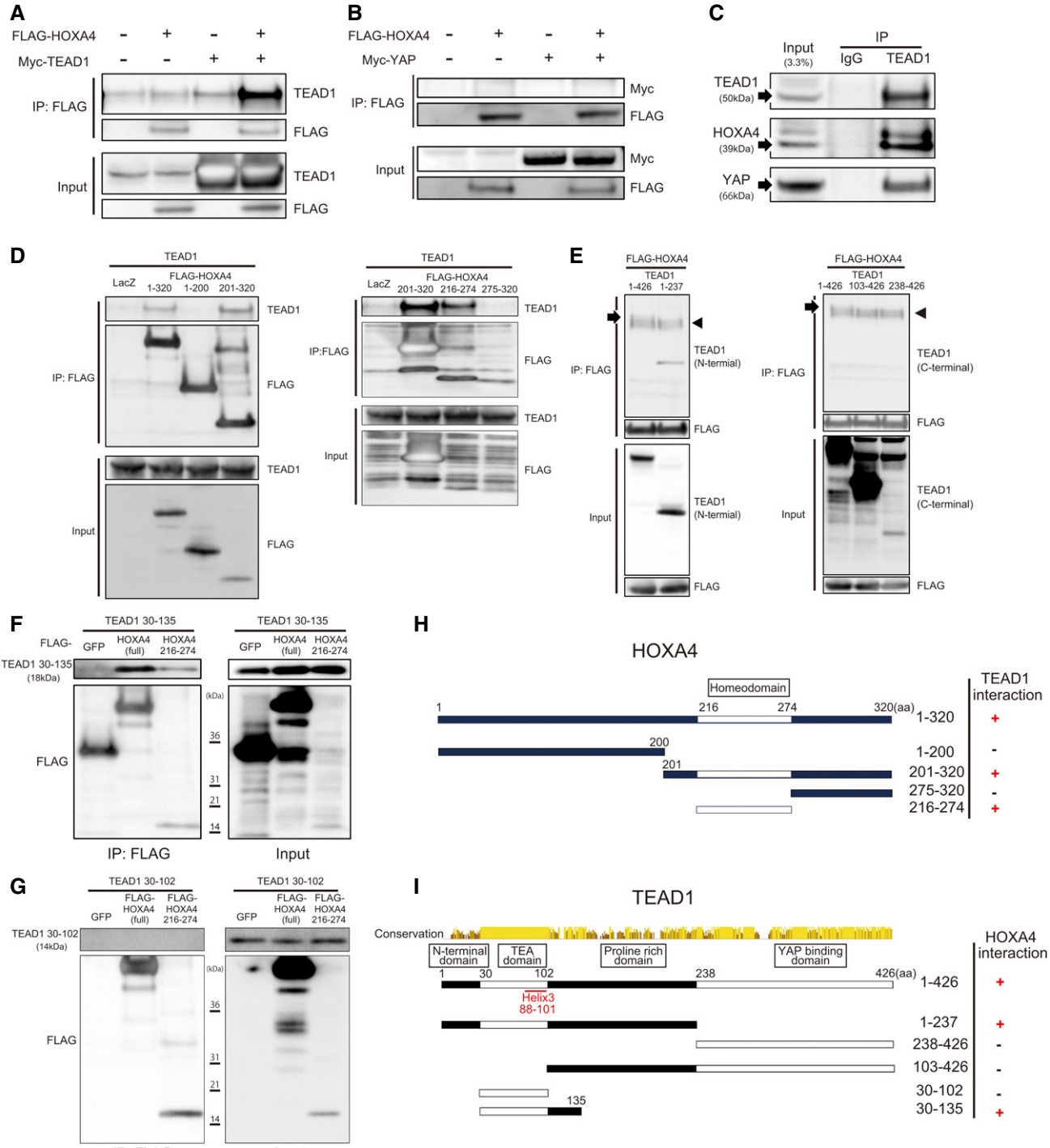

**Figure 3.**

**Figure 3.   Protein–protein interactions between HoxA4 and TEADs.**

A, B   Association of HOXA4 with TEAD1 (A) but not YAP (B) indicated by co-IP experiments. HEK 293T cells were co-transfected with FLAG-HOXA4 and Myc-TEAD1 or Myc-YAP.

C   Interaction between endogenous TEAD1 and HOXA4 by co-IP experiments in HEK 293T cells. The interaction between endogenous TEAD1 and YAP was used as a positive control.

D   Immunoprecipitation using FLAG antibody in lysates from HEK 293T cells transfected with plasmids expressing HOXA4 truncation mutants and TEAD1.

E   Immunoprecipitation using FLAG antibody in lysates from HEK 293T cells transfected with plasmids expressing TEAD1 truncation mutants and HOXA4. Arrowheads indicate IgG heavy chain. Arrows indicate TEAD1 (a.a. 1–426).

F   Requirement of TEAD1 (a.a. 30–135) for its interaction with HOXA4 via its homeodomain (a.a. 216–274).

G   Failure of TEAD1 (a.a. 30–102), TEA domain alone, to interact with HOXA4.

H, I   Diagram showing the structure of truncated HOXA4 (H) and TEAD1 (I) constructs and possible protein–protein interactions between them. Conservation between human TEAD1, TEAD2, TEAD3, and TEAD4 is shown (I).

Data information: All experiments above were repeated at least twice.

the other is HOXA4 might attenuate TEAD-DNA-binding affinity. To verify theses hypothesis, we performed several chromatin immuno-precipitation (ChIP) assays in HEK 293T cells and assessed whether HOXA4 affects the binding of YAP or TEAD4 to the target chromatin at the CTGF and CYR61 promoters [29]. Both TEAD4 and YAP occupation to these promoters were decreased by HOXA4, but disruption of YAP binding was predominant rather than TEAD4 (Fig 4A). Of note, this blocking effect on YAP was more remarkable with high YAP activity (i.e., low cell density) than low YAP activity (i.e., high cell density). Moreover, a competitive co-IP assay in HEK 293T cells showed HOXA4 and nuclear-localized form of YAP (YAP-S127A) competed with each other for binding to TEAD4 (Fig 4B). Together, these results provide compelling evidence that HOXA4 competes with YAP for binding to TEADs to inhibit the expression of YAP/TEAD target genes (Fig 4C).

## HOXA4 specifically represses YAP/TEAD transcriptional activity among HOX families

From the result of the shRNA screen, activation of YAP/TEAD-dependent transcription was highly specific for knockdown of HOXA4 compared with the knockdown of other HOX genes included in the library (Fig EV3A). However, because HOX family members have strong sequence conservation in the homeodomain, we wondered whether other HOX genes could interact with TEADs. Although HOXA9 is linked to the susceptibility to atherosclerosis [22], HOXA9 did not change YAP-5SA-induced 8×GTIIC-luciferase activity (Fig EV3B) or CTGF expression (Fig EV3C). Furthermore, only a modest decline in CTGF and CYR61 expression induced by YAP-5SA was observed with the truncated form of HOXA4 including the homeodomain (Fig EV3D). Together, these data indicate non-conserved regions other than the homeodomain of HOXA4 are essential for the inhibition of YAP/TEAD-dependent transcription.

## HOXA4 maintains the differentiation state in vascular smooth muscle cells by inhibiting YAP/TEAD-mediated phenotype switching

To investigate HOXA4 function *in vivo*, we first examined the expression pattern of *Hoxa4* in mice. *Hoxa4* was highly expressed in lung, white adipose tissue, brown adipose tissue, carotid artery, and aorta (Fig 5A). There have been already some reports showing HOXA4 is related to lung cancer development [30] or adipocyte differentiation [31], but the vascular function of HOXA4 is largely unknown. FANTOM5 human gene expression datasets show high

expression levels of *HOXA4* in VSMCs as well as fetal organs (Fig EV4A) [32,33]. We also confirmed the expression of HOXA4 in a human aortic tissue (Fig EV4B). VSMC phenotypic switching from a differentiated contractile phenotype to a synthetic proliferative phenotype is a hallmark of vascular diseases. Recently, it has been reported that the expression of YAP is induced during VSMC phenotypic transformation, and YAP plays a critical role in both promoting VSMC proliferation and inhibiting smooth muscle-specific gene expression [3,4]. Therefore, we hypothesized that HOXA4 might be involved in the maintenance of VSMC differentiation via YAP/TEAD suppression.

In response to PDGF-BB, a growth factor known to induce VSMC transformation [34], the expression of *HOXA4* was significantly downregulated in cultured aortic SMCs with a reduction in differentiated VSMC marker gene expression both in human (Fig 5B) and in mouse (Fig EV4C). This finding was consistent with previous findings of reduced *HOXA4* expression in human VSMCs in response to other inflammation stimuli, interferon gamma, and tumor necrosis factor-alpha [35]. Similar to other cell types, HOXA4 and TEAD were mainly localized in the nucleus, whereas YAP and TAZ were located in both the nucleus and cytoplasm in human VSMCs (Figs EV4D and 4E).

YAP in human primary aortic smooth muscle cells attenuated the expression of smooth muscle contractile genes, including *α-SMA*, *SM22α*, *Myh11*, and *calponin* in transcription levels as described previously (Fig 5C). On the contrary, HOXA4 facilitated the expression of these smooth muscle contractile genes (Fig 5C). On the other hand, mRNA expression of YAP/TEAD target genes, such as *CTGF*, *CYR61*, transforming growth factor-beta 2 (*TGFB2*), [36] and cyclin D1 (*CCND1*) [4], was attenuated by HOXA4 (Fig 5F). We confirmed these changes of differentiated VSMC marker genes using the protein levels by shYAP/TAZ (Fig 5D) and shHOXA4 (Fig 5E). Furthermore, overexpression of HOXA4 significantly attenuated VSMC proliferation (Figs 5G and EV4E), although silencing of endogenous HOXA4 resulted in enhanced rates of VSMC growth (Figs 5I and EV4F). This negative effect of HOXA4 on the proliferative capacity of VSMCs was also confirmed by bromodeoxyuridine (BrdU) incorporation analysis (Fig 5H and J). These phenotypic changes induced by HOXA4 were completely the opposite to those by YAP.

Such opposing phenotypic effects in VSMCs induced by HOXA4 and YAP demonstrated that YAP/TEAD might be negatively regulated by HOXA4. To investigate whether YAP/TEAD is involved in HOXA4-mediated inhibition of VSMC phenotype switching, we performed knockdown experiments of TEAD or YAP combined with

                                                                                                 

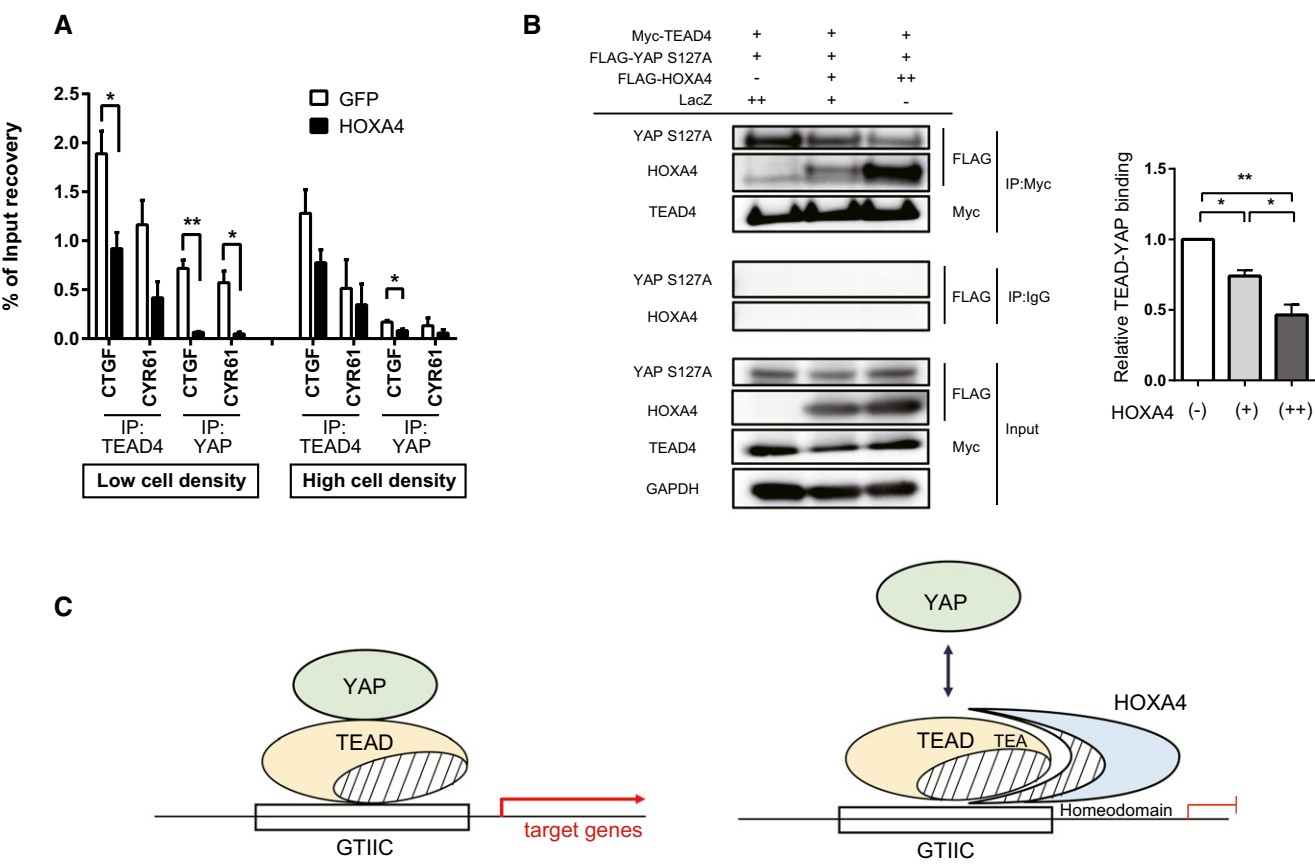

**Figure 4. HOXA4 negatively regulates TEAD-YAP interaction.**

A  ChIP assays of CTGF and CYR61 promoters using TEAD4 or YAP antibodies in HEK 293T cells transfected with plasmids expressing HOXA4. *P < 0.05, **P < 0.01, by unpaired two-tailed Student's t-test. Data are expressed as the means ± SEM of three independent experiments.

B  Competitive co-IP assay and densitometry showing significantly reduced TEAD-YAP interaction by overexpression of HOXA4 in a dose-dependent manner in HEK 293T cells. *P < 0.05, **P < 0.01, by ANOVA with Sidak's correction. Data are presented as mean ± SEM of three independent experiments.

C  Working hypothesis whereby HOXA4 inhibits YAP/TEAD-mediated transcription activity.

HOXA4. As well as previous investigation of rat VSMCs [5], TEAD1 is predominantly expressed among the four isoforms (Appendix Fig S4) in human VSMCs; then, we utilized shRNA targeting common sequence in TEAD1, TEAD3, and TEAD4. Reduced smooth muscle-specific protein by knockdown of HOXA4 was rescued by concomitant TEAD knockdown (Fig 6A), and YAP knockdown also canceled the increased proliferation rate by HOXA4 abrogation (Fig 6B). Disruption of the TEAD-HOXA4 interaction in VSMCs by knockdown of HOXA4 significantly increased the amount of endogenous TEAD-YAP complexes (Fig 6C) and increased the occupation of HOXA4 on the TEAD-binding region in the promoters of *CTGF* and *CYR61* but not *α-SMA* attenuated that of phosphorylation-defective YAP (Fig 6D). Together, these data demonstrate that HOXA4 has a potent negative effect on VSMC phenotypic switching via YAP/TEAD suppression in the same manner identified in HEK 293T.

**Loss of HOXA4 exacerbates vascular remodeling in mice**

To clarify the pathophysiological function of HOXA4 in VSMCs, we created a *Hoxa4*-deficient mouse line using CRISPR-Cas9 gene editing (Appendix Fig S5A) [37]. The gene length of *Hoxa4* gene is 1.6 kilobase pairs and contains only two exons. In order to exclude the possibility that a functional truncated peptide was translated, we deleted almost all genomic regions of *Hoxa4* by inducing double-strand breaks at two sites around the start and stop codons and following homologous recombination repair (Appendix Fig S5B and Fig 5C). YAP/TEAD is important for organ size control, but there was no significant difference in organ weight between *Hoxa4* KO and WT adult mice (Appendix Fig S6A). A staining for VSMCs and endothelial cells of *Hoxa4* KO embryos detected no morphological difference in the dorsal aorta compared with WT (Appendix Fig S6B). The contractibility of primary VSMCs harvested from *Hoxa4* KO and WT mice was also compared using a collagen gel assay and found to be almost similar (Appendix Fig S6B).

At baseline, there were very slight differences in smooth muscle marker gene expression in the carotid artery; however, YAP/TEAD transcriptional target genes related to cell growth are significantly upregulated in *Hoxa4* KO mice compared with WT mice (Fig 7A and D). The mouse carotid artery ligation model is commonly used for investigating vascular remodeling due to proliferation of phenotypically switched VSMCs [38]. Indeed, mice with genetic deletion of YAP in VSMCs attenuate neointima formation in the proximal

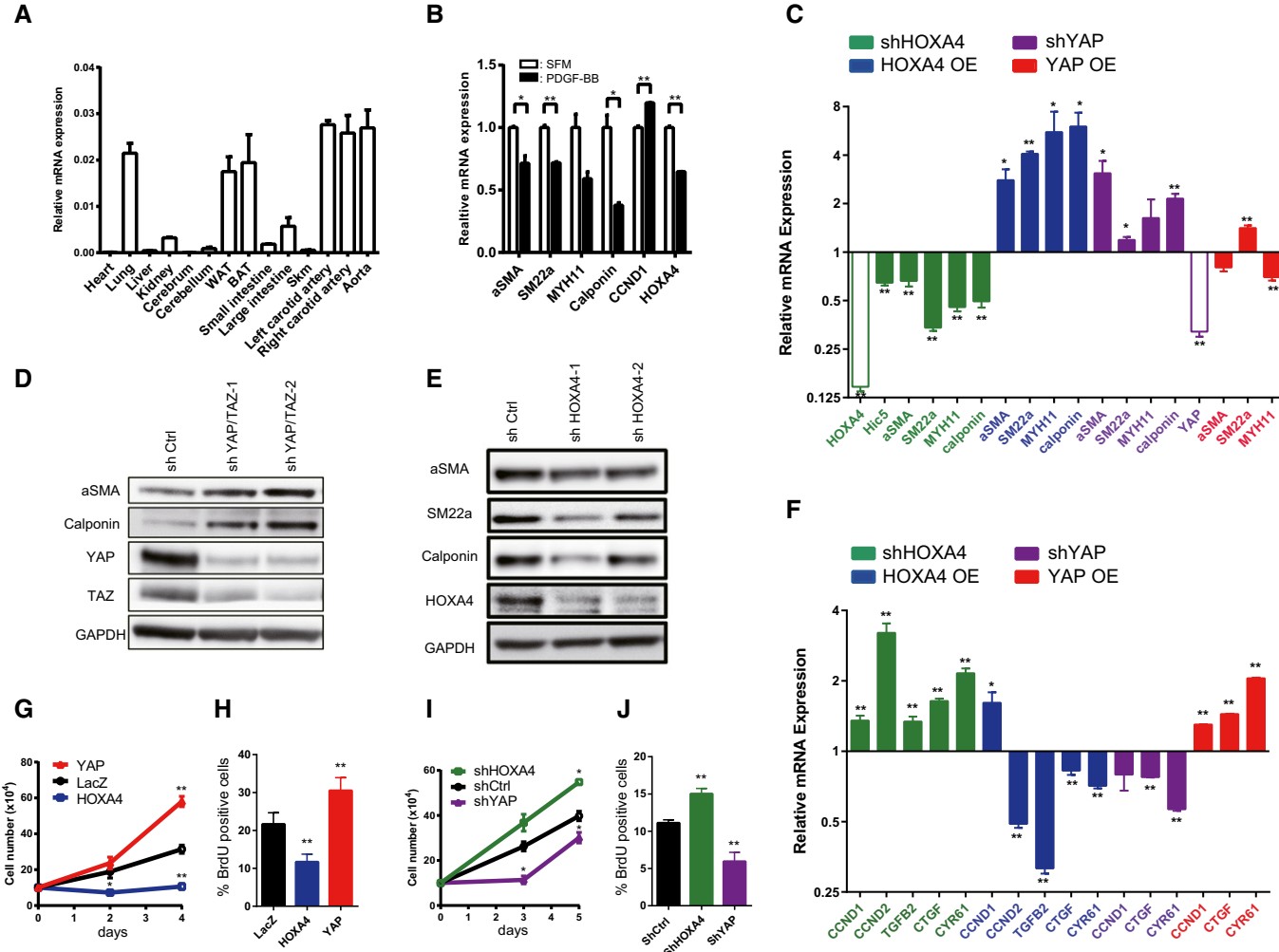

**Figure 5. HOXA4 regulates the phenotypic switching of vascular smooth muscle cells.**

A  Expression of *Hoxa4* in various mouse tissues using qRT–PCR. Expression of glyceraldehyde-3-phosphate dehydrogenase (*Gapdh*) mRNA was used as an internal control. WAT; white adipose tissue, BAT; brown adipose tissue, Skm; skeletal muscle.

B  Quantitative real-time PCR analysis of differentiated VSMC marker genes in primary human VSMCs with PDGF-BB stimulation for 24 h. *$P < 0.05$, **$P < 0.01$, by unpaired two-tailed Student's *t*-test.

C  Quantitative real-time PCR analysis of smooth muscle-specific genes in primary human VSMCs transduced with lentiviral vectors encoding overexpression or knockdown of HOXA4 and YAP. Cells infected with lentiviral vectors encoding LacZ or shCtrl were used as controls. *$P < 0.05$, **$P < 0.01$, by unpaired two-tailed Student's *t*-test.

D, E  Representative Western blotting analysis of total protein lysates of primary human VSMCs transduced with lentiviral vectors encoding knockdown of YAP/TAZ (D) or HOXA4 (E). Representative Western blotting analysis of total protein lysates of primary human VSMCs transduced with lentiviral vectors encoding knockdown of HOXA4.

F  Quantitative real-time PCR analysis of YAP/TEAD target genes in primary human VSMCs transduced with lentiviral vectors encoding overexpression or knockdown of HOXA4 and YAP. The cells infected with lentiviral vectors encoding LacZ or shCtrl were used as controls. *$P < 0.05$, **$P < 0.01$, by unpaired two-tailed Student's *t*-test.

G  Proliferation assay in VSMCs transduced with the indicated genes. Virus-transfected human VSMCs were plated at equal density in growth medium, and then, cells were collected and counted at each time point as indicated. **$P < 0.01$ versus LacZ, by ANOVA with Sidak's correction.

H  BrdU incorporation assays in VSMCs transduced with the indicated genes. The percentage of BrdU-positive cells in each group is shown. Nine to twelve images were randomly acquired and analyzed. **$P < 0.01$ versus LacZ, by ANOVA with Sidak's correction.

I  Proliferation assay in VSMCs transduced with the indicated genes. Virus-transfected human VSMCs were plated at equal density in growth medium, and then, cells were collected and counted at each time point as indicated. *$P < 0.05$ versus shCtrl, by ANOVA with Sidak's correction.

J  BrdU incorporation assays in VSMCs transduced with the indicated genes. The percentage of BrdU-positive cells in each group is shown. Nine to twelve images were randomly acquired and analyzed. **$P < 0.01$ versus shCtrl, by ANOVA with Sidak's correction.

Data information: Graphs are presented as means ± SEM; three biological repeats, except for (H) and (J).

portion of the ligated carotid artery [3], so we assessed vascular remodeling of *Hoxa4* KO mice with this experimental model. Down-regulations of differentiated smooth muscle genes are usually observed in 1 week after ligation, much earlier than morphological remodeling occurs [5,38,39]. A more remarkable reduction in smooth muscle marker gene expression was observed in *Hoxa4* KO

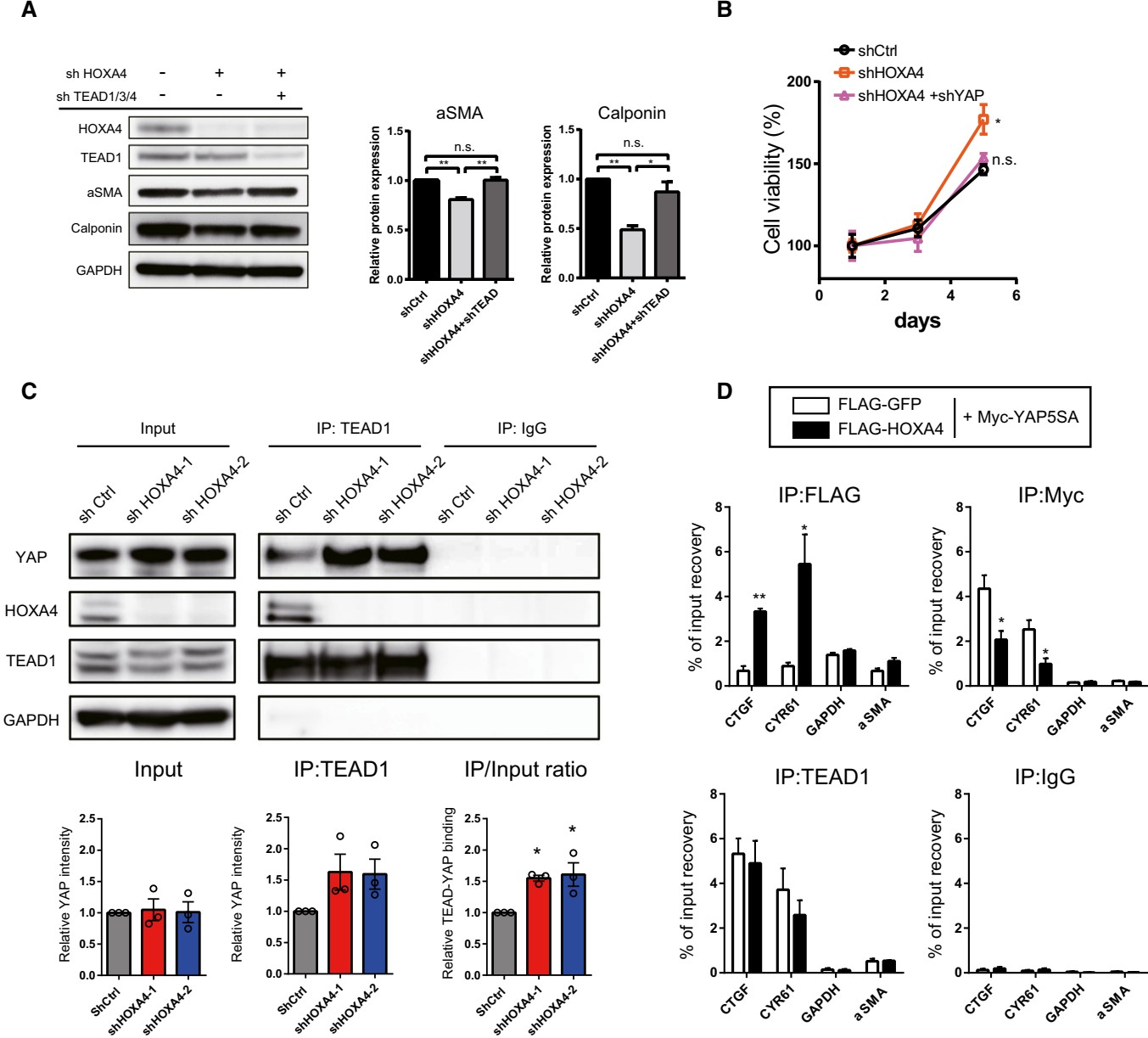

**Figure 6. A negative effect of HOXA4 on VSMC phenotypic switching is mediated by YAP/TEAD.**

A  Representative Western blotting analysis of total protein lysates of primary human VSMCs transduced with lentiviral vectors encoding shRNAs of HOXA4 and TEAD1/3/4. Quantification of alpha-smooth muscle actin and calponin protein levels is expressed as the means ± SEM of three independent experiments. *P < 0.05, **P < 0.01, by ANOVA with Sidak's correction.

B  Proliferation assay in VSMCs transduced with indicated genes. Virus-transfected human VSMCs were plated at equal density in growth medium, and then, cell viability was analyzed by Cell Counting Kit-8 at indicated days at each time point as indicated. *P < 0.05 versus shCtrl, by ANOVA with Sidak's correction. Data are presented as mean ± SEM of three independent experiments.

C  Representative immunoblotting images and quantification of the interaction between endogenous TEAD1 and HOXA4 or YAP using co-IP experiments in human VSMCs with knockdown of HOXA4. *P < 0.05 versus shCtrl, by ANOVA with Sidak's correction. Data are presented as mean ± SEM of three independent experiments.

D  ChIP assays of CTGF, CYR61, and α-SMA promoters using anti-FLAG-tag, anti-Myc-tag, and anti-TEAD1 antibodies in human VSMCs transfected with plasmids expressing Myc-tagged YAP-5SA and/or FLAG-tagged HOXA4. *P < 0.05, **P < 0.01 versus FLAG-GFP, by unpaired two-tailed Student's t-test. Data are expressed as the means ± SEM of three independent experiments.

mice compared with WT mice (Fig 7B). Although we could not detect a specific signal of mouse *Hoxa4* protein due to the absence of commercially available antibodies (See Materials and Methods), the transcription of *Hoxa4* in the ligated carotid artery was

significantly reduced at 1 week after operation (Fig 7C). The predominant expression of YAP/TEAD target genes in the carotid arteries of *Hoxa4* KO mice was also observed at 1 week after ligation (Fig 7E). These differences in differentiation- or proliferation-

associated genes in the injured artery between WT and *Hoax4* KO mice were also confirmed by immunoblotting (Fig 7F) or immuno-histochemistry (Fig EV5A), leading to a twofold increase in neointima formation in *Hoxa4* KO mice at 250 μm proximal to the ligation (Fig 7G–I) and at 500 μm proximal to the ligation (Figs EV5B and 5D). Taken together, the above data demonstrate that Hoxa4 prevents VSMC phenotypic transformation and following vascular remodeling after arterial ligation injury *in vivo*. Mechanistically, such phenotype is, at least in part, due to YAP/TEAD suppression via TEAD-HOXA4 interaction as observed from *in vitro* results.

## Discussion

Here, by using a pooled shRNA screen, we demonstrated that HOXA4 is a novel negative modulator of YAP/TEAD transcriptional activity and also suppressed YAP/TEAD-induced phenotypic switching of VSMCs *in vitro* and *in vivo*. In spite of numerous discoveries of regulators of YAP/TEAD-mediated transcription that work upstream of the Hippo signaling cascade, few modulators have been detected to affect downstream molecular events. We first performed an unbiased screen using a pooled shRNA library and detected several candidate genes of putative YAP/TEAD repressors. Although the library does not include well-known negative regulators of YAP/TEAD such as cell surface proteins, shYAP-transduced cells presented the lowest GFP intensity, and shRNAs targeting MAPK [40], ERBB4 [16], and Wnt [41] signaling pathway-related genes were accumulated in the "GFP low" population. These pathways are known to have a positive effect on YAP/TEAD transcriptional activity, which validated our initial screen. Meanwhile, in the "GFP high" population, shRNAs targeting ribosome biogenesis- or translation-related genes were enriched. These processes are fundamental biological functions of cells and affect many signals. Therefore, we excluded these genes as candidates.

HOXA4 is one of the HOX genes, which consist of 39 genes clustered into four chromosomes in human and mouse. Each HOX gene encodes a conserved DNA-binding motif, homeodomain of 61 amino acids, with spatial and temporal expression patterns during the embryonic period, where they work as a transcription factor and regulate the expression of downstream genes to build segments or structures in the body [21]. Therefore, mutations in HOX genes cause congenital disorders [42]. Recent studies have provided increasing evidence that HOX genes continue to be regionally expressed even in the adult organisms and related to disease formation and development including cardiovascular diseases. For instance, HOXA9 has an athero-protective effect via inhibiting NF-kB-dependent gene expression in endothelial cells and VSMCs [22]. HOXA5 is also an athero-protective gene that suppresses blood flow-dependent endothelial inflammation [23].

To date, the molecular function of HOXA4 in adult organs is largely unknown. However, HOXA4 has been reported as a tumor suppressor [43], and a recent study revealed its involvement in lung cancer progression via inhibiting the Wnt-β-catenin signaling by upregulation of GSK3β [30]. In fact, whole-genome sequencing of young patients with non-small-cell lung cancer detects mutations of HOXA4 as well as those of MST1, the upstream kinase of the Hippo signaling pathway, as cancer-associated gene mutations [44]. Although the molecular mechanisms of HOXA4-mediated

regulation of cell proliferation may be different among each cell type, Wnt signaling and Hippo signaling interact closely with each other [41,45]; then, further investigation is needed to test the possible cross-talk between these two pathways in relation to HOXA4.

Mechanistically, HOXA4 inhibits YAP/TEAD transcriptional activity via HOXA4-TEADs interaction. Because the HOXA4 interaction region in TEAD1 (residues 30–135) overlaps with its DNA recognition helix (residues 88–101) required for DNA binding of TEAD1 (Fig 3F) [46], we first considered the possibility HOXA4-TEAD1 interaction might affect the DNA-binding ability of TEAD1. However, HOXA4 had little effect on the DNA-TEAD affinity unexpectedly, whereas a much more remarkable reduction in TEAD-mediated YAP-DNA binding was observed (Figs 4A and 6D). Then, co-IP assays confirmed that HOXA4 competes with YAP for TEAD binding. This finding is a reasonable explanation that the inhibition of YAP/TEAD transcriptional activity was not seen in other HOX genes in spite of their binding abilities to TEAD via the homeodomain. Together, these data indicated that non-conserved regions other than the homeodomain of HOXA4 mediate the inhibition of TEAD-YAP interactions. Recently, several molecules have been shown to suppress YAP/TEAD transcriptional activity via interaction with TEAD, such as VGLL-4 [26], RUNX3 [28], and HNF4α [27]. However, the mechanisms of transcriptional suppression are completely different among these molecules including HOXA4. VGLL4 is a transcriptional cofactor without a DNA-binding domain. The interaction region of TEADs with the TDU domain of VGLL4 overlaps the YAP-binding domain of TEADs, which inhibits the YAP-TEAD interaction [26]. On the other hand, RUNX3 and HNF4α are transcription factors. RUNX3 binds to both TEAD4 and YAP, forming ternary complex of RUNX3-TEAD4-YAP. The interaction domain of TEAD4 with the Runt domain of RUNX3 overlaps with the DNA recognition helix (H3), and thus, RUNX3 abrogates the ability of TEAD4 to bind DNA, whereas RUNX3 does not affect the YAP-TEAD4 interaction [28]. HNF4α is associated with TEAD4 but not YAP. HNF4α competes with YAP to interact with TEAD4, resulting in a reduction in YAP-DNA binding just like HOXA4. However, HNF4α associates with both, but rather the N-terminal than the C-terminal regions of TEAD4, and does not change TEAD-DNA affinity [27], which are different from HOXA4.

In our study, *Hoxa4* KO mice show no apparent phenotype other than the previously reported mild skeletal alterations at baseline [47]. A possible explanation of this finding is that other Hox genes have functional redundancy in terms of development, which is supported by a previous report of embryonic lethality observed in *Hoxa4/Hoxb4* double-mutant mice [48]. On the other hand, we found *Hoxa4* KO mice showed exacerbated neointima formation of the ligated carotid artery, in which YAP/TEAD transcriptional activity is highly upregulated (Fig 7). This observation implies that HOXA4 has a specific molecular function and may be important for the maintenance of homeostasis in YAP-activating conditions, such as cancer [18] or vascular remodeling [3], because repression of the YAP-TEAD interaction by HOXA4 was only observed with high YAP activity (i.e., low cell density) (Fig 4A).

YAP/TEAD plays a critical role in the phenotypic switching of VSMCs and subsequent vascular remodeling [3–5]. YAP/TEAD is also activated by well-known VSMC phenotypic switch-inducing factors such as PDGF-BB [4], angiotensin II [49], and thromboxane

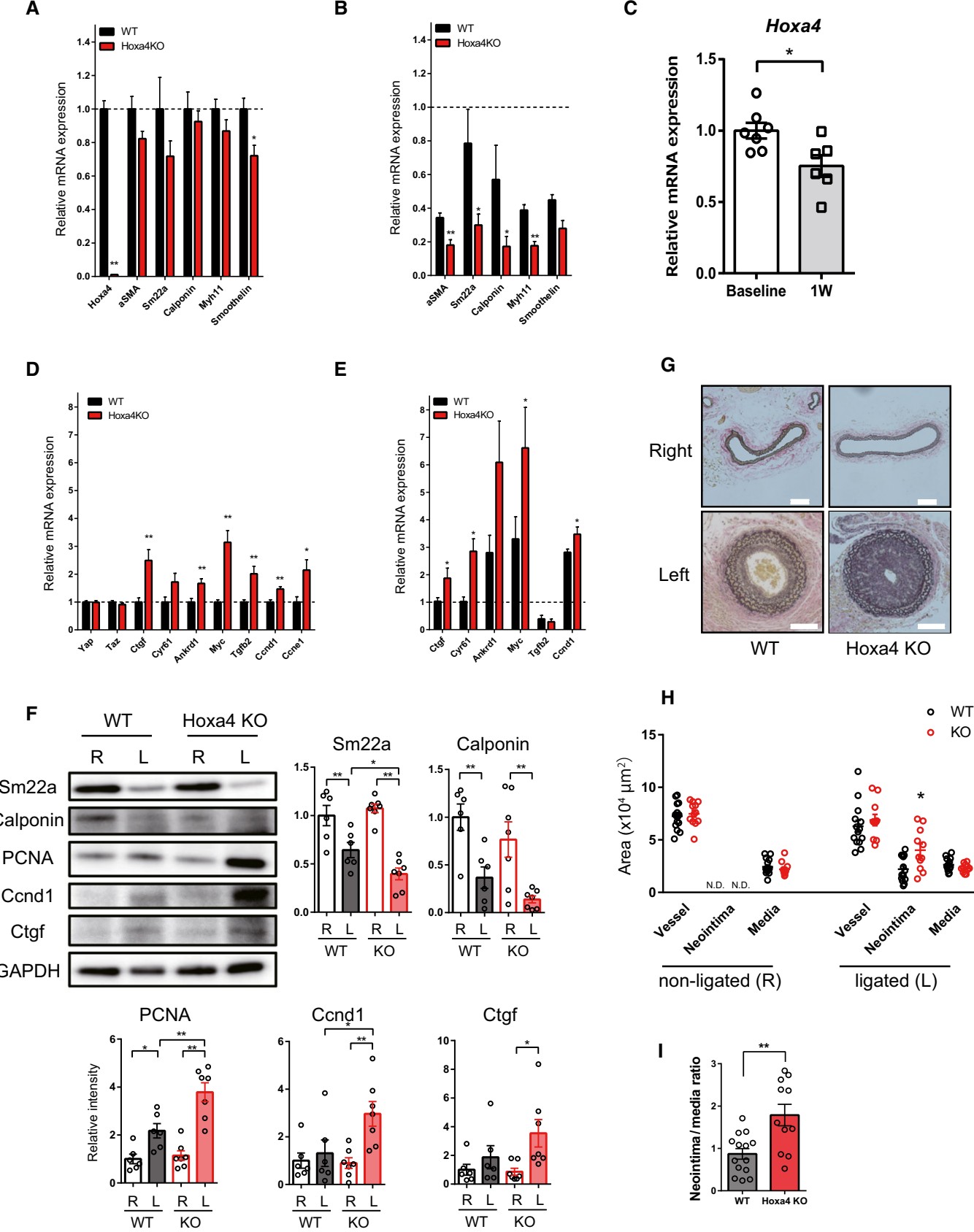

**Figure 7.**

**Figure 7.** *Hoxa4* KO mice alleviate smooth muscle phenotypic modulation and abrogate neointima formation after carotid artery ligation injury.

A Quantitative real-time PCR analysis of differentiated smooth muscle marker genes in the left carotid artery of 8-week-old WT and *Hoxa4* KO mice (WT: *n* = 7; *Hoxa4* KO: *n* = 5).

B Quantitative real-time PCR analysis of differentiated smooth muscle marker genes in the ligated left carotid artery of WT and *Hoxa4* KO mice at 1 week after ligation (*n* = 6). Mean expression level of six WT mice at the baseline was defined as 1.0.

C *Hoxa4* expression levels in the ligated left carotid artery of WT mice at 1 week after ligation (*n* = 6).

D Quantitative real-time PCR analysis of Yap/Tead target genes in the left carotid artery of 8-week-old WT and *Hoxa4* KO mice (WT: *n* = 7; *Hoxa4* KO: *n* = 5).

E Quantitative real-time PCR analysis of Yap/Tead target genes in the ligated left carotid artery of WT and *Hoxa4* KO mice at 1 week after ligation (*n* = 6). Mean expression level of six WT mice at the baseline was defined as 1.0.

F Representative Western blotting analysis and densitometry of non-ligated (right; R) and ligated (left; L) carotid arteries from WT and *Hoxa4* KO mice at 1 week after ligation (WT: *n* = 6; *Hoxa4* KO: *n* = 7). *$P < 0.05$, **$P < 0.01$, by two-way ANOVA with Sidak's correction.

G–I Representative images showing Elastica van Gieson staining (G); vessel, neointima, and media area (H); and neointima-to-media layer ratio (I) of carotid arteries of WT and *Hoxa4* KO mice at 4 weeks after ligation (WT: *n* = 14; *Hoxa4* KO: *n* = 11).

Data information: Images and data were obtained at 250 μm proximal to the ligation. Scale bars indicate 100 μm. *$P < 0.05$, **$P < 0.01$ versus WT, except in (C) versus baseline, by unpaired two-tailed Mann–Whitney test. Data are presented as mean ± SEM. N.D., not detected.

A2 [50]. Inhibition of YAP/TEAD activity alleviates VSMC phenotypic transformation *in vitro* and vascular remodeling *in vivo*. Although upstream regulators of YAP/TEAD, such as Rho-actin signaling followed by a change in YAP phosphorylation, are well-investigated [10,36,50], the downstream molecules to affect YAP/TEAD activity in VSMCs remain to be elucidated. Previous observations that downregulation of smooth muscle contractile genes is induced by YAP but not YAP-S94A (TEAD-binding-defective mutant) [5] and that upregulation of proliferative genes such as *CTGF, CYR61, ANKRD1, Myc, TGFβ2*, and *CCND1* is inhibited by verteporfin (an inhibitor of the binding of YAP and TEAD) [36,49] indicate phenotypic switching of VSMCs is mediated by YAP/TEAD-dependent transcription. In this study, we detected HOXA4 as a novel downstream repressor of YAP/TEAD transcriptional activity in VSMCs and maintenance of HOXA4 expression could be a potential therapeutic strategy for vascular remodeling.

There were several limitations to this study. First, the function of HOXA4 as a transcription factor is still unknown; therefore, the phenotype of human VSMCs or Hoxa4 KO mice in our study might be partially affected by HOXA4 target genes. Second, it is uncertain whether the phenotype of exacerbating neointima formation in *Hoxa4* KO mice is due to only synthetic VSMCs or in conjunction with other cell types such as endothelial cells. These questions are to be investigated further.

We conclude that HOXA4 maintains the differentiation of VSMCs by inhibiting YAP/TEAD-induced VSMC phenotypic switching. These findings give us a better understanding of the vascular pathophysiology and a novel therapeutic approach for vascular remodeling.

# Materials and Methods

### Plasmids

pENTR-3×FLAG-YAP-S127A (#42239) [51], pCMV-Flag-YAP-5SA/S94A (#33103) [14], 8×GTIIC-luciferase (#34615) [10], 3×Flag-pCMV5-TOPO TAZ-S89A (#24815) [52], human HOXA9-MSCV-short (FLAG-tagged) (#20978) [53], Myc-TEAD4 (#24638) [54], and pCMX-Gal4-TEAD2 (#33107) [14] were obtained from Addgene. A wild-type (WT) Yes-associated protein (YAP) construct was created from a YAP-S127A construct using the QuikChange II Site-Directed

Mutagenesis Kit (Agilent Technologies) in accordance with the manufacturer's instructions. A YAP-5SA construct was created from the YAP-5SA/S94A construct, and a WT transcriptional coactivator with PDZ-binding motif (TAZ) construct was created from the TAZ-S89A construct in the same way. Because large tumor suppressor kinase (LATS)-mediated phosphorylation of serine is critical for the inactivation of YAP in the cytoplasm, YAP $^{serine\ 127→alanine}$ (S127A) or YAP $^{serine\ 61,\ 109,\ 127,\ 164,\ 381→alanine}$ (5SA) mutants are constitutive active form of YAP, which remain in the nucleus [9].

Protein-coding regions of human TEA domain transcription factor 1 (TEAD1), TEAD3, and Homeobox A4 (HOXA4) were subcloned from cDNA of HEK 293T cells. All constructs were correctly inserted into a lentiviral vector driven by a cytomegalovirus (CMV) promoter with 3×FLAG-tag or Myc-tag at the N-terminus.

### Generation of 8×GTIIC-EmGFP-293T cell line

An 8×GTIIC-luciferase plasmid containing eight TEAD-binding sites (CATTCCA) was obtained from Addgene (#34615) [10], and its luciferase sequence was changed to emerald green fluorescent protein (EmGFP) subcloned from pcDNA™6.2-GW/EmGFP-miR vector (Invitrogen). Then, 8×GTIIC-minimal promoter-EmGFP sequence was subcloned and introduced into a lentiviral vector without a promoter. After lentiviral transduction into HEK 293T cells, forty clones were obtained. Among them, a single clone that has the highest EmGFP reactivity to YAP-S127A, shLATS2, and cell density was isolated and named as 8×GTIIC-EmGFP 293T cell line.

### ShRNA screening

Packaging for pooled lentiviral shRNA library (DECIPHER™ Human Module 1; Cellecta) was performed in accordance with the manufacturer's instructions, and lentiviral particles were made. The library consists of 27,500 shRNAs targeting 5,043 genes with a puromycin-resistance marker and expression of tagged red fluorescent protein (RFP) fluorescent protein as transduction indicators. A total number of $8 × 10^6$ 8×GTIIC-EmGFP 293T cells were infected with the pooled lentiviral shRNA library at a multiplicity of infection (MOI) of 0.3 to ensure that only one shRNA was introduced into each cell. The transduced cells were

selected for 5 days with puromycin (1 μg/ml) with maintenance of a library coverage of more than 1,000-fold. After cells were maintained overconfluent for more than 48 h to ensure that EmGFP intensity was maximally suppressed, cells were sorted based on RFP using a fluorescence-activated cell sorter (FACS) Aria™ III (BD Biosciences). Then, cells were sorted into two subpopulations, "GFP high" or "GFP low", representing 5 or 50% of the cells with highest or lowest GFP signals, respectively. Genomic DNA was extracted from a total of $5.5 \times 10^5$ cells in the "GFP high" population and $2.8 \times 10^7$ cells in the "GFP low" population, and shRNA-specific barcodes were PCR-amplified from genomic DNA using HT Sequencing Kits (Illumina) in accordance with the manufacturer's instructions. Both populations were then analyzed by Illumina sequencing, and the frequency of each shRNA was determined using Barcode Deconvolution software (Cellecta). After quantile normalization, the $\log_2$ (lead abundance of top 5% GFP-expressing population/that of bottom 50%) was detected.

## Cell lines and transfection

HEK 293T cells were purchased from the American Type Culture Collection. Lipofection was used for DNA transduction to HEK 293T cells. Transfection of overexpression or shRNA-mediated knockdown vectors into HEK 293T cells was performed using polyethylenimine "MAX" (Polysciences); then, cells were used for experiments 2 or 3 days later.

## Dual-luciferase assay

A total of $4 \times 10^4$ HEK 293T cells were seeded in 24-well plates and transfected with 0.4 μg of expression or knockdown vectors and 0.2 μg of 8×GTIIC-luciferase with 0.005 μg of *Renilla* luciferase reporter by lipofection. Three days later, both luciferase activities were measured using a dual-luciferase reporter assay system (PicaGene dual kit; Toyo Ink).

## Human and mouse primary vascular smooth muscle cells

Primary human aortic smooth muscle cells (Kurabo) were cultured in growth medium provided by the manufacturer and used for experiments at passage 5 (P5).

Mouse aortic smooth muscle cells were isolated from thoracic aortas of 8-week-old male mice as described previously [55]. Briefly, the adventitia and endothelium were removed after digestion with collagenase type II (175 units/ml; Worthington) for 20 min. Subsequently, the media were digested with collagenase type II (175 units/ml) and elastase (0.5 mg/ml; Sigma) for 45–60 min. Isolated cells were cultured on collagen type I-coated dishes (IWAKI) in Dulbecco's modified Eagle's medium (DMEM) supplemented with 10% fetal bovine serum (FBS), 100 units/ml penicillin, and 100 μg/ml streptomycin at 37°C with 5% $CO_2$, and then, cells were used for experiments at passage 5 (P5).

For PDGF-BB stimulation, cells were seeded in 6-well plates at a concentration of $2.5 \times 10^5$ cells per well and serum-starved with differentiation medium provided by the manufacturer or DMEM containing 0.5% FBS for 24 h, and then subsequently stimulated with 10 ng/ml PDGF-BB (WAKO) for 24 h.

## Lentivirus transfection

Lentiviral transfection was used for DNA transduction into human VSMCs. Lentiviral stocks were produced in HEK 293T cells as described previously [56–58]. In brief, virus-containing medium was collected 48 h post-transfection and filtered through a 0.45-μm filter. One round of lentiviral infection was performed by replacing the medium with virus-containing medium (containing 8 μg/ml of polybrene), followed by centrifugation at 1,220 *g* for 30 min at 32°C. Cells were used for analysis 3 days after DNA transduction. For knockdown assays, lentiviral vectors coding shRNA designed by BLOCK-iT™ RNAi Designer (Invitrogen) were used. The siRNA target sequences are as follows [14]: siControl, AAATGTACTGCG CGTGGAG; siHOXA4-1, CCAAGATGCGATCCTCCAA; siHOXA4-2, G CAGAAGAAGACAGACCCT; siLATS1, GCAGCTGCCAGACCTATTA; siLATS2, GGAAGATCCTCTACCAGAA; siSAV1, GGAAGATCCTCTA CCAGAA; siYAP-1, GACATCTTCTGGTCAGAGA; siYAP-2, CTGGT CAGAGATACTTCTT; siTAZ-1, CCTCAATGGAGGGCCATAT; and siTAZ-2, CCGTTTCCCTGATTTCCTT.

## Quantitative real-time PCR

Total RNA was isolated using TriPure Isolation Reagent (Sigma-Aldrich) from cells or tissues of mice using a homogenizer. Single-strand cDNA was synthesized from 100 to 1,000 ng of total RNA using a Verso cDNA Synthesis Kit (Thermo Fisher), and the products were analyzed using a thermal cycler (StepOnePlus; Applied Biosystems) with THUNDERBIRD® SYBR qPCR Mix (TOYOBO). Samples were normalized using β-actin mRNA expression in experiments with HEK 293T cells or GAPDH mRNA expression in experiments with vascular smooth muscle cells or mouse tissues.

## Protein extraction and Western blotting

Cultured cells or carotid arteries of mice were homogenized in lysis buffer consisting of 100 mM Tris–HCl, 75 mM NaCl, and 1% Triton X-100 at pH 7.4 (Nacalai Tesque) supplemented with cOmplete Mini Protease Inhibitor (Roche), 0.5 mM NaF, and 10 μM $NaVO_4$ just prior to use. Cytoplasmic and nuclear protein fractions were separated using NE-PER™ (Thermo Fisher) in accordance with the manufacturer's instructions. A total of 20–30 μg (*in vitro*) or 5 μg (*in vivo*) of protein was fractionated using NuPAGE 4–12% Bis-Tris gels (Invitrogen) and transferred to a nitrocellulose membrane (Whatman). The membrane was blocked using phosphate-buffered saline (PBS) containing 5% non-fat milk for 30 min or Blocking One-P (Nacalai Tesque) for 20 min and incubated with a primary antibody overnight at 4°C. After a washing step in PBS-0.05% Tween-20, the membrane was incubated with a secondary antibody for 1 h at room temperature. The membrane was then washed and detected using Pierce™ ECL or ECL Plus Western Blotting Substrate (GE Healthcare) with a LAS-4000 (Fuji Film). Quantification of Western blots was performed using ImageJ64 software (NIH). Antibodies used are as follows: anti-TEADs (directed against the N-terminus of TEADs; 1:1,000; CST #13295), anti-FLAG (1:1,000; MBL PM020), anti-FLAG (1:1,000; Sigma-Aldrich F1804), anti-Myc-tag (1:1,000; CST #2278), anti-Myc-tag (1:5,000; Sigma-Aldrich M4439), anti-HOXA4 (1:1,000; Abcam ab131049), anti-YAP (1:1,000; CST #14074), anti-TEAD1 (directed against the C-terminus of TEAD1;

1:5,000; Abcam ab109080), anti-α-SMA (1:1,000; Sigma-Aldrich C6198), anti-SM22α (1:400; Santa Cruz sc-53932), anti-TAZ (1:500; Sigma-Aldrich HPA007415), anti-YAP/TAZ (1:200; Santa Cruz sc-101199), anti-calponin (1:200; Santa Cruz sc-58707), anti-PCNA (1:500; Santa Cruz sc-56), anti-cyclin D1 (1:1,000; Abcam ab92566), anti-CTGF (1:200; Santa Cruz sc-365970), anti-GAPDH (1:3,000; CST #2118), anti-lamin A/C (1:400; Santa Cruz sc-6215), anti-rabbit IgG, HRP-linked (1:2,000; GE NA934V), anti-mouse IgG, HRP-linked (1:2,000, GE NA931V), and anti-goat IgG, HRP-linked (1:2,000, Abcam ab6885).

## Co-immunoprecipitation assay

A total of $2.0$–$3.0 \times 10^6$ HEK 293T cells or human VSMCs were used for co-immunoprecipitation assays. The total cell lysate was incubated with the indicated antibodies and Dynabeads Protein G (Thermo Fisher) or anti-FLAG antibody-conjugated magnetic beads (WAKO) in accordance with the manufacturer's instructions. Samples were extracted from magnetic beads using 2× LDS (twofold dilution of NuPAGE™ 4× LDS Sample Buffer; Thermo Fisher) by boiling for 5 min at 95°C. The immune complexes were subjected to SDS–PAGE and analyzed by Western blotting. Antibodies used are as follows: anti-TEAD1 (2 µg; Santa Cruz sc-376113), normal mouse IgG1 (2 µg, Santa Cruz sc-3877), anti-Myc-tag (1:200, CST #2278), and anti-FLAG (antibody-conjugated beads; WAKO 017-25151).

## Chromatin immunoprecipitation

A total of $8.0 \times 10^6$ HEK 293T cells or human VSMCs were used for chromatin immunoprecipitation assays. For cross-linking, medium was replaced with 4.85 ml of DMEM supplemented with 140 µl of formaldehyde (FA) and incubated at room temperature for 10 min with rotation. After the medium was replaced with 5 ml of glycine solution (0.125 M), cells were suspended with cold PBS and centrifuged. The supernatant was discarded, and the cross-linked samples were homogenized in the hypotonic lysis buffer using a Dounce homogenizer supplemented with protease inhibitor cocktail, nuclease inhibitor, DTT, and NP40. The samples were then centrifuged at 10,500 *g* for 20 min at 4°C. The pellet was resuspended with immunoprecipitation buffer supplemented with protease inhibitor cocktail, nuclease inhibitor, dithiothreitol (DTT), and phenylmethylsulfonyl fluoride (PMSF), and chromatin was sheared by sonication (30-s on and 30-s off, 10 cycles) using a BIORUPTOR® (Cosmo Bio). After the samples were centrifuged at 15,300 *g* for 10 min at 4°C, the supernatant (nuclear fraction) was divided and then antibodies for TEAD4 (1–2 µg; Abcam ab58310), YAP (1–2 µg; Abcam ab52771), TEAD1 (2 µg; Santa Cruz sc-376113), normal mouse IgG1 (2 µg, Santa Cruz sc-3877), FLAG-tag (2 µg; Sigma-Aldrich F1804), or Myc-tag (2 µg; Sigma-Aldrich M4439) were added. The mixtures of antibody and the nuclear extract were incubated overnight at 4°C with rotation and then added to Dynabeads Protein G and incubated for 2 h at 4°C with rotation. One-tenth of the nuclear extract was retained as the input sample. After incubation, the supernatant was removed, and the beads were washed three times with immunoprecipitation buffer supplemented with the protease inhibitor cocktail, RNase inhibitor, DTT, and PMSF. The beads were then treated with DNA elution buffer supplemented with RNase H (New England Biolabs) and RNase A (Sigma-Aldrich) for 30 min at 37°C twice,

followed by the treatment with proteinase K (Sigma-Aldrich) for 1 h at 65°C. After this, DNA was extracted and quantification of TEAD-binding sequences in *CTGF*, *CYR61* [29], or *α-SMA* [59] promoter was performed using quantitative real-time PCR. Primers used for ChIP assays are as follows [29]: human *CTGF* promoter: forward, GCCAATGAGCTGAATGGAGT; reverse, CAATCCGGTGTGAGTTG ATG; human *CYR61* promoter: forward, CCCTTGGCTGTTATGAG GAA; reverse, CCTTGCATTCCTTTGCATTT.; and human *α-SMA* promoter: forward, GGGACCTCAGCACAAAACTC; reverse, GAAG GCTTGGCGTGTTTATC. A pair of primer for human *GAPDH* was used as a negative control [60]: forward, CCACATCGCTCAGAC ACCAT; reverse, CCCGCAAGGCTCGTAGAC.

## Databases

Gene expression data in human various cell types or organs are obtained from RefEx (Reference Expression dataset: http://refex.dbc ls.jp/) [32], an information browsing tool for cap analysis gene expression (CAGE) data from the RIKEN FANTOM5 project (http://fantom.gsc.riken.jp/5) [33].

## Immunocytochemistry

Human VSMCs or HEK 293T cells expressing HOXA4-GFP fusion protein were plated in a Nunc® Lab-Tek® II CC2™ Chamber Slide™ (Sigma-Aldrich) and then fixed in 4% paraformaldehyde (PFA) for 20 min at room temperature on the next day, followed by three washes in PBS. Then, cells were permeabilized in 0.1% Triton for 10 min, followed by three washes in PBS. Subsequently, the cells were incubated in blocking buffer (5% donkey serum in PBS) for 15 min and then incubated overnight at 4°C with primary antibodies in blocking buffer. After three washes in PBS, cells were incubated with secondary antibodies and DAPI (1 µg/ml) for 1 h at room temperature. The slides were rinsed with PBS and mounted. Immunofluorescence was detected using an Axio Observer 7 (Zeiss). Primary antibodies used were as follows: anti-HOXA4 (1:100; Abcam ab131049), anti-GFP (1:50; Santa Cruz sc-9996), and anti-YAP (1:100; CST #14074).

## Cell viability assay

For proliferation assays, cells were trypsinized at 2 days after lentiviral infection and plated on 60-mm dish (5 ml) or in a 96-well plate (100 µl) at a concentration of $2 \times 10^4$ cells/ml. Cell numbers were counted using Countess II FL (Thermo Fisher), and cell viability was measured using Cell Counting Kit-8 (Dojindo) at the indicated time points.

## BrdU incorporation assay

*In vitro* bromodeoxyuridine (BrdU) incorporation was performed to analyze cell proliferation. Human VSMCs were plated in a Nunc® Lab-Tek® II CC2™ Chamber Slide™ (Sigma-Aldrich) at a density of $1.0 \times 10^4$ cells/cm² at 2 days after transduction of indicated lentiviral vectors. The lentivirus-containing medium was diluted twice for overexpression experiments due to high cell toxicity compared with the knockdown experiments. Another 2 days later, cells were stimulated with 10 µM BrdU (Sigma, B9285) for 4 h and fixed with 4%

paraformaldehyde (PFA) for 30 min at 4°C. The slides were auto-claved (90°C, 20 min) in HistoVT One (Nacalai Tesque) for heat-mediated antigen retrieval. Then, cells were permeabilized in 1.0% Triton for 15 min and blocked in 5% donkey serum/PBS for 15 min, followed by incubation with an anti-BrdU antibody (Abcam, ab6326, 1:250 dilution) at 4°C overnight. The slides were rinsed three times and then incubated with donkey anti-rat IgG secondary antibody Alexa Fluor 594 (1:200; Thermo Fisher A11007) with DAPI (0.5 µg/ml) at room temperature for 1 h. The slides were washed and mounted in VECTASHIELD® Mounting Medium (Vector Labora-tories, H-1000) and then observed using an Axio Observer 7 (Zeiss) with a 10× objective. Images were randomly acquired, and the percentage of BrdU-positive cells in each image was counted using ImageJ64 for analysis.

### Collagen gel contraction assay

Collagen gel contraction assays were performed using a Collagen-based Cell Contraction Assay Kit (Funakoshi Co., Ltd.) in accor-dance with the manufacturer's instructions. Briefly, primary aortic smooth muscle cells were harvested from 8-week-old male mice. A concentration of $4 \times 10^5$ cells/ml cell–collagen mixture was placed in a 24-well plate in a volume of 500 µl. After gelation for 1 h at 37°C, 1.0 ml of culture medium was added and the plate incubated for 48 h. After the gels were gently released from each well, the size of the gel was sequentially photographed at the indicated time points and analyzed using ImageJ64 software.

### Human samples of aorta

Human aortic samples were obtained from patients with abdominal aortic aneurysm (AAA) who underwent the artificial blood vessel replacement surgery with prior informed consent and written permission. Tissues were immediately fixed with 4% PFA for 48 h and embedded in paraffin.

### Mice

Cas9 mRNA and single-guide RNAs (sgRNAs) were synthesized *in vitro* using a Transfection-ready Cas9 SmartNuclease mRNA Kit (System Biosciences) and a Linearized T7 gRNA SmartNuclease Vector Kit (System Biosciences) in accordance with the manufac-turer's instructions. The following sequences were used for sgRNA synthesis: left sgRNA, GAACTTGGGCTCGATGTAGT; right sgRNA, CTCCATATAATCTAGAGACC.

*Hoxa4* knockout (KO) mice were generated using the CRISPR/Cas9 system, as described previously [37]. In brief, Cas9 mRNA and sgRNAs were microinjected into fertilized embryos of C57BL/6J mice to induce subsequent non-homologous end joining. Mutations in the *Hoxa4* allele were confirmed by Sanger sequencing analyses (ABI 3100 Genetic Analyzer). All mice were genotyped within 4 weeks after birth, using PCR with specific primers as follows: mouse *Hoxa4* forward, ACGACACCGCGAGAAAAATTAG; mouse *Hoxa4* reverse, CTGTGCCCCAGTATAAGACAAC. Homozygous KO mice were generated from a heterozygous intercross, and WT litter-mates were used as a control. All *in vivo* experiments were conducted using 8-week-old male mice with a C57BL/J background, except for measurement of body/organ weight (9-week-old male

mice) and immunohistochemistry of embryos (embryonic day [E] 16.5). E0.5 was defined as the day that vaginal plug was detected.

Mice were maintained in specific pathogen-free conditions at the Institute of Laboratory Animals of Kyoto University Graduate School of Medicine.

### Carotid artery ligation model

Carotid artery ligation was performed as described previously [3,38]. In brief, mice were anesthetized with a mixture of medeto-midine, midazolam, and butorphanol; then, the left common carotid artery was exposed and completely ligated just proximal to the carotid bifurcation with 7-0 silk suture. Then, carotid arteries were harvested at 7 days after surgery for the extraction of RNA and protein or 28 days after surgery for histological analysis.

### Histological analysis and immunohistochemistry

For the analysis of mouse carotid arteries, mice were perfused from the left ventricle with PBS followed by 4% PFA. The injured (left) and contralateral uninjured (right) arteries were dissected, fixed with 4% PFA, and embedded in paraffin. Cross sections of the bilat-eral carotid arteries were stained with hematoxylin and eosin (HE) and Elastica van Gieson (EVG). Neointima formation as well as vessel and media area was assessed at two different portions, 250 and 500 µm proximal to the ligature, using ImageJ64 software (NIH). Sections were also obtained from injured carotid arteries at 1,000 µm for immunohistochemistry; then, the samples were immunolabeled with an Alexa Fluor 488-conjugated anti-PCNA anti-body (Abcam, ab201672, 1:50 dilution) and an anti-CTGF antibody (Abcam, ab6992, 1:50 dilution) followed by anti-rabbit IgG secondary antibody Alexa Fluor 594 and DAPI.

For the analysis of mouse embryos, embryos were euthanized by decapitation, fixed with 4% PFA for 24 h and an additional 48 h after the trunk was cut at the level between neck and the root of forearms, and then embedded in paraffin. Cross sections of dorsal aorta were immunolabeled with a Cy3-conjugated anti-α-SMA anti-body (1:200; Sigma-Aldrich C6198) and an anti-CD31 antibody (1:50; Abcam ab28364) followed by anti-rabbit IgG secondary anti-body Alexa Fluor 488 and DAPI.

For the analysis of human tissue, AAA samples were immunola-beled with a Cy3-conjugated anti-α-SMA antibody (1:200; Sigma-Aldrich C6198) and an anti-HOXA4 antibody (1:100; Abcam ab28364) followed by anti-rabbit IgG secondary antibody Alexa Fluor 488 and DAPI. Images were acquired from the relatively healthy, marginal zone of AAA. Antigen retrieval using HistoVT One and blocking using Blocking One Histo (Nacalai Tesque) were performed in accordance with the manufacturer's instructions after deparaffinization in all studies described above.

### Antibodies

Although all commercially available antibodies (Abcam ab131049, Abcam ab26097, Santa Cruz sc-398426, Santa Cruz sc-515418, MyBioSource MBS9206995, MyBioSource MBS2525887, and Biorbyt orb157564) that are described as being applicable for immunoblot-ting and/or immunohistochemistry of mouse origin in their data-sheets were tried to detect mouse *Hoxa4*, no specific signal was

observed in comparisons between WT and *Hoxa4*-deficient mice-derived samples. For the detection of human HOXA4, Abcam ab131049 was validated for immunoblotting or immunohistochemistry by knockdown and overexpression of HOXA4 in HEK 293T cells and human VSMCs.

## Statistical analysis

Data are presented as means ± standard error of the mean (SEM). For statistical comparisons, unpaired Student's *t*-test (two groups, parametric), Mann–Whitney test (two groups, non-parametric), one-way analysis of variance (ANOVA) with Sidak's post hoc test (three or more groups, parametric), or two-way ANOVA with Sidak's post hoc test (two different categorical variables, parametric) was used as indicated in the figure legends. A $P < 0.05$ was considered statistically significant. Statistical analyses were performed using GraphPad Prism 6 (GraphPad Software, Inc.).

## Study approval

All animal experiments were approved by Kyoto University Ethics Review Board and Animal Research Committee of Kyoto University. The study using human samples was approved by Institutional Review Board/Ethics Committee of the Kyoto University Graduate School of Medicine (No. G473) and conducted under the tenets of the Declaration of Helsinki and the Department of Health and Human Services Belmont Report.

Expanded View for this article is available online.

## Acknowledgements

performed at the Medical Research Support Center, Graduate School of Medicine, Kyoto University. This work was supported by grants from the Ministry of Education, Culture, Sports, Science and Technology (MEXT) and Japan Society for the Promotion of Science (JSPS) KAKENHI Grant Numbers 17H04177, 17H05599, and 17K19590 to K. Ono, 17K09860 to T. Horie, and JP1605297 to T. Kimura, and a visioning research grant (Step) from Takeda Science Foundation to K. Ono.

## Author contributions

KO and MK designed experiments. MK performed animal experiments with the help of TH, YI, ST, and YM. MK and OB performed FACS studies. MK performed other *in vitro* experiments with the help of RRR, TW, TY, CO, and SX. MK, YN, TK, and KO analyzed the data and prepared the manuscript. All authors discussed and commented on the manuscript.

## Conflict of interest

The authors declare that they have no conflict of interest.

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
