## [Review Process File · EMBO Reports]

Homeobox A4 Suppresses Vascular Remodeling by Repressing YAP/TEAD Transcriptional Activity

Masahiro Kimura, Takahiro Horie, Osamu Baba, Yuya Ide, Shuhei Tsuji, Randolph Ruiz Rodriguez, Toshimitsu Watanabe, Tomohiro Yamasaki, Chiharu Otani, Sijia Xu, Yui Miyasaka, Yasuhiro Nakashima, Takeshi Kimura, Koh Ono

Review timeline:

Submission date:	1 May 2019
Editorial Decision:	5 June 2019
Revision received:	5 December 2019
Editorial Decision:	22 January 2020
Revision received:	31 January 2020
Accepted:	12 February 2020

Editor: Deniz Senyilmaz-Tiebe

Transaction Report:

1st Editorial Decision

5 June 2019

Thank you for submitting your manuscript for consideration by EMBO Reports. It has now been seen by two referees.

As you can see, both referees express interest in the proposed role of HOXA4 in regulation of YAP/TEAD mediated vascular remodeling. However, they also raise concerns that need to be addressed in full before we can consider publication of the manuscript here.

Given these constructive comments, I would like to invite you to revise your manuscript with the understanding that the referee must be fully addressed and their suggestions taken on board. Please address all referee concerns in a complete point-by-point response. Acceptance of the manuscript will depend on a positive outcome of a second round of review. It is EMBO Reports policy to allow a single round of revision only and acceptance or rejection of the manuscript will therefore depend on the completeness of your responses included in the next, final version of the manuscript.

REFeree REPORTS**Referee #1:**

The manuscript by Masahiro Kimura and colleagues describes HoxA4- initially identified in an unbiased shRNA screen- as a putative repressor of Yap/Taz-TEAD transcriptional activity in

vascular remodeling. In *in vitro* experiments in HEK293T and primary VSMC and in *in vivo* mouse arterial injury models, the authors provide evidence that HoxA suppression promotes cellular proliferation, YT-TEAD target gene activation and exacerbates vascular remodeling after carotid artery ligation. Vice versa, loss of HoxA4 or YAP-TEAD activation inhibits contractile smooth muscle marker gene expression.

Hox genes encode a subset of the homeobox transcription factors that are known to control the specification of anteroposterior identity in the animal embryo. They are also known to regulate stem cell differentiation and are usually dysregulated in cancer. Whereas the general role of YAP in phenotypic switching in VSMC has been suggested before, the negative regulation of this process by Hox4a and the proposed competitive inhibition of YAP binding to TEAD are to this reviewer's knowledge novel and worth reporting in EMBO reports. Having said that, the manuscript although shedding light on HoxA as a putative negative regulator of TEAD signaling, falls short in comprehensive experimental preparation and final conclusions on vascular remodeling in its current form. The following points should be addressed before publication:

- The list/table of shRNAs that repress 8xGTIC-EmGFP expression in different conditions "low and high density" and to which extend should be provided. They stated that they excluded shRNAs targeting ribosome biogenesis or transcription related genes that were enriched in the GFP high population, on the other hand they decided to have a closer look onto the function of HoxA4.

- Does HoxA4 have an effect on GTIC or target gene expression in the presence of Yap-S94A (nuclear YAP lacking TEAD binding)?

- Fig.3: title and conclusion too far-fetched

"HoxA4 associates with TEADs to inhibit YAP-mediated transcriptional activation"

The authors have shown immunoprecipitation studies of truncated TEAD1 and HoxA4 to map the minimal TEAD-HoxA4 interaction domain but lack experiments showing which truncations block YAP-TEAD complex formation and GTIC-dependent repression

- Fig.4A: CHIP experiments for SMA, SM22a, Calponin and YT-target (e.g. Cyr61, CTGF) promoters using TEAD1/2, YAP and HoxA4 antibodies should be included preferably in VSMCs (TEAD4 is not expressed in VSMC)

- Fig.4B: The authors stated that YAP and HoxA compete for binding to TEADs to inhibit YT target gene expression. They provided CHIP assay that point towards this conclusion, while the data for the competitive CoIP are not conclusive since also the amount of YAP-S127A in the input is reduced to the same extend. The question of whether shHoxA4 increases TEAD-YAP interaction needs also to be addressed. Quantifications are required and the IgG control in this particular experiment is missing.

In general input controls for immunoprecipitation experiments throughout the manuscript lack Actin or Gapdh as loading control.

- Fig.7: HoxA4-KO mouse and the arterial injury model

Phenotype of HoxA4-ko mice: is there overgrowth in some organs where YT-TEAD signature is increased (eg lungs) and are other Hox family members overtaking the function of HoxA4 in HoxA4-KO?

Immunofluorescence analysis, WB as well as qRT PCRs in carotid arteries of ctr and HoxA4-KO before and after ligation (or Right-nonligated, left-ligated) should be provided to show the increase in YT-target gene expression and/or nuclear localization of YAP/TAZ before and after ligation.

- EVF2D: higher magnification needs to be provided as well as YAP and DAPI staining to show where HoxA4 and YAP localize in the different conditions

- EVF5D: the author state that "VSMCs with nuclear HoxA4 were spindle shaped like a contractile phenotype, whereas cells without nuclear HoxA4 were rhomboid shaped like synthetic phenotype" (P12, lines 4-6). First, this is not obvious by the provided magnification it could be also dependent on the confluency. Second, which cells are those, i.e. under which conditions HoxA4 can be found in the cytoplasm and does HoxA4 have a function there?

Referee #2:

The manuscript entitled 'Homeobox A4 suppresses vascular remodelling as a novel regulator of YAP/TEAD transcriptional activity' by Kimura et al., describes the role of HoxA4 as a novel suppressor of YAP/TEAD transcriptional activity as well as its role as a suppressor in the phenotypic switching of VSMCs. The authors have performed a thorough investigation to elucidate the molecular function of HOXA4 with proper validation. The findings are interesting for the field of YAP/TEAD and transcriptional biology. However, no evidence of the interaction of HOXA4 and TEADs leading to attenuation of YAP activity is shown in VSMCs. Furthermore, a better characterization of the HOXA4 KO mice should be provided.

Major comments:

1. The key mechanistic findings of the study should be shown in VSMCs as well and not only in HEK293T.
2. Figure 2: the authors claim that the interaction between HOXA4 and TEAD would play a role during vascular remodeling; however, this interaction should be shown in VSMCs as well; and without overexpression of TEAD and HOXA4.
3. Figure 5G,H: counting the cell number does not really tell whether cells proliferate more or less, as other mechanisms such as cell death could be taking place. Therefore, this analysis should be replaced by a better proliferation assay (i.e. like BrdU labelling).
4. Figure 5 (extended view): HOXA4 expression should be shown in vivo via immunostaining on tissue sections and not just based on the FANTOM5 CAGE data.
5. Figure 7: A better characterization of the HOXA4 KO mouse and in-vivo analysis needs to be shown. A staining for vSMCs and ECs would be a nice approach to show in general how vessels and SMCs look like in control and mutant embryos.
6. Could the contractibility of vSMC in control and HOXA4 mutants be measured? For example, in aortic rings?
7. Figure 7B : Complementary to Panel B: In order to link comprehensively the mechanism of how HOXA4 displaces YAP, does vascular remodeling induce HOXA4 expression? which would then displace YAP in vivo, as seen in the competition assay in vitro? A staining for HOXA4 could complement the RNA expression levels after artery ligation.
8. Figure 7D:
 - Include H&E and/or Elastica van Gieson images from the four groups: right non-ligated artery WT animals; left ligated artery WT animals; right non-ligated artery HOXA4KO; left ligated artery HOXA4KO to have a better idea on how the neointima looks in every condition with the same staining.
 - Include analysis of neointima area and neointima/media ratio for the four groups mentioned before.
 - Complement the morphometric analysis with the other vessel parameters: vessel area and media area between the four groups.

Minor Comments:

1. A little work on wording and structure would make manuscript sound much better.
2. Figure 3 and figure 4B, no 'n' number is provided.
3. The authors have used student T-test in most of their analysis (comparison between 2 different variables), however, Mann whitney U test (non-parametric) was used in figure 7B. Is there a special reason for this?
4. The authors mention that "HOXA4 in vivo seems to be dispensable in steady-state conditions in adults". However, this is not shown in their study. Has this been properly investigated in VSMCs in vivo in other studies? Could the authors elaborate this more in discussion and give the appropriate references?

5. Figure 4B: It doesn't look like a dose response when expressing different levels of HOXA4. Quantification of WB should be provided.
6. Figure 5E: contrast in aSMA and Calponin bands seems very high and artificial. Improve blot.
7. Typo on expanded view figure 4, panel B: should say fold CHANGE
8. Typo on expanded view figure 5, panel B: should say RELATIVE
9. Figure 2 C and D: a one-way ANOVA was performed to determine statistical differences among groups however it is not stated whether control group (YAP5SA4 -, HoxA4-) has a significant difference with nuclear YAP overexpression (YAP5SA4 +, HoxA4-), especially in the analysis of the YAP target genes CTGF and CYR61.
10. A clearer labeling of Figure7 panels is recommended

1st Revision - authors' response

5 December 2019

Response to Referee #1

We are grateful to Referee #1 for the informative and useful comments. As described below, we have considered all of these comments and used them to improve our manuscript.

The manuscript by Masahiro Kimura and colleagues describes HoxA4- initially identified in an unbiased shRNA screen- as a putative repressor of Yap/Taz-TEAD transcriptional activity in vascular remodeling. In in vitro experiments in HEK293T and primary VSMC and in in vivo mouse arterial injury models, the authors provide evidence that HoxA suppression promotes cellular proliferation, YT-TEAD target gene activation and exacerbates vascular remodeling after carotid artery ligation. Vice versa, loss of HoxA4 or YAP-TEAD activation inhibits contractile smooth muscle marker gene expression.

Hox genes encode a subset of the homeobox transcription factors that are known to control the specification of anteroposterior identity in the animal embryo. They are also known to regulate stem cell differentiation and are usually dysregulated in cancer. Whereas the general role of YAP in phenotypic switching in VSMC has been suggested before, the negative regulation of this process by Hox4a and the proposed competitive inhibition of YAP binding to TEAD are to this reviewers knowledge novel and worth reporting in EMBO reports. Having said that, the manuscript although shedding light on HoxA as a putative negative regulator of TEAD signaling, falls short in comprehensive experimental preparation and final conclusions on vascular remodeling in its current form. The following points should be addressed before publication:

We thank the reviewer for the valuable and constructive comments on our manuscript. We have followed the suggestions and believe that these changes have considerably improved our manuscript

1. The list/table of shRNAs that repress 8xGTIIC-EmGFP expression in different conditions "low and high density" and to which extend should be provided. They stated that they excluded shRNAs targeting ribosome biogenesis or transcription related genes that were enriched in the GFP high population, on the other hand they decided to have a closer look onto the function of HoxA4.

As the reviewer suggested, we are also interested in shRNAs that repress TEAD-mediated transcriptional activity. Cells transduced with shRNAs that repressed 8xGTIIC-EmGFP expression were considered to be accumulated in the population representing the lowest GFP signal. To address this comment, we should compare cells with lowest and higher GFP signals, e.g. bottom 1–5% vs top 50%, and the cells should be kept sparse during screening because GFP intensity is easily suppressed by cell–cell contact, meanwhile a large number of cells are needed for high library complexity. However, we initially aimed to discover a novel YAP/TEAD inhibitor and compared only two subpopulations, the bottom 50% vs top 5% GFP intensity and due to the methodological

difficulty described above, it is impossible to show shRNAs that repressed 8×GTIIC-EmGFP expression from our current experiments.

From our initial screen, we detected several shRNAs targeting ribosomes or translation complexes such as Eukaryotic initiation factor (EIF), Ribosomal protein large subunit (RPL), Ribosomal protein small subunit (RPS). We corrected the manuscript as "translation-related genes."

Modified sentence (on page 16, lines 13-14):

Meanwhile, in the "GFP high" population, shRNAs targeting ribosome biogenesis- or transcription translation-related genes were enriched.

2. Does HoxA4 have an effect on GTIIC or target gene expression in the presence of Yap-S94A (nuclear YAP lacking TEAD binding)?

We assessed the expression of CTGF and CYR61 with co-transfection of YAP-5SA-S94A and HOXA4 in HEK 293T cells. As described previously (Zhao B. et.al., Genes Dev. 2008; 22(14):1962-71), YAP-5SA-S94A induced only modest upregulation of these genes expression (7-fold and 3.5-fold, respectively) compared with YAP-5SA (30-fold and 13-fold, respectively) and the inhibition effect by HOXA4 was also slight (-30%) compared to YAP-5SA (-70%). We think the inhibition effect of HOXA4 is more remarkable in situations of increased YAP-TEAD binding.

3. Fig.3: title and conclusion too far-fetched.

"HoxA4 associates with TEADs to inhibit YAP-mediated transcriptional activation"

The authors have shown immunoprecipitation studies of truncated TEAD1 and HoxA4 to map the minimal TEAD-HoxA4 interaction domain but lack experiments showing which truncations block YAP-Tead complex formation and GTIIC-dependent repression.

We agree with the reviewer's comment. We modified the title as "Protein-protein interactions between HoxA4 and TEADs". To address this comment, we assessed the expression of target genes induced by YAP-5SA with each HOXA4-truncated mutant. The mutants that retained TEAD-binding homeodomains, HOXA4 201-320 or 216-272, significantly but only modestly suppressed their expression, implying a region other than the structural TEAD-binding sites of HOXA4 are necessary for YAP inhibition (Fig EV 3D). We added Figure EV3D and an explanation on page 10, lines 18-20, and modified the title of Figure 3.

Inserted sentence (on page 13, lines 3-7):

Furthermore, only a modest decline in CTGF and CYR61 expression induced by YAP-5SA was observed with the truncated form of HOXA4 including the homeodomain (Fig EV3D).

Modified title (on page 46, lines 10):

Figure 3 - HOXA4 associates with TEADs to inhibit YAP-mediated transcriptional activation
Protein-protein interactions between HoxA4 and TEADs.

4. Fig.4A: CHIP experiments for SMA, SM22a, Calponin and YT-target (e.g. Cyr61, CTGF) promoters using TEAD1/2, YAP and HoxA4 antibodies should be included preferably in VSMCs (TEAD4 is not expressed in VSMC).

We thank the reviewer for this essential and important comment. We performed ChIP assays using human vascular smooth muscle cells with the expression of tagged-HOXA4 and YAP-5SA (a nuclear-localized mutant), because we have no commercially available antibody against HOXA4 for ChIP, and YAP-DNA binding via TEADs is largely affected by YAP subcellular localization. As shown in Fig 6D, the occupation of YAP in the promoter of CTGF and CYR61 was also inhibited by HOXA4 in vascular smooth muscle cells, whereas TEAD-DNA binding was unchanged. On the other hand, both YAP and TEAD1 did not accumulate in α -SMA promoter, and there are no TEAD-binding consensus sequences in the promoters of SM22a and Calponin. This means that YAP-TEAD-mediated repression of smooth muscle contractile genes could not be due to direct transcriptional regulation and the mechanisms remain to be elucidated in future investigations. We added Figure 6D and an explanation on page 13, lines 3-7.

Inserted sentence (on page 13, lines 3-7):

Disruption of the TEAD-HOXA4 interaction in VSMCs by knockdown of HOXA4 significantly increased the amount of endogenous TEAD-YAP complexes (Fig 6C) and increased the occupation of HOXA4 on the TEAD-binding region in the promoters of CTGF and CYR61 but not α SMA attenuated that of phosphorylation-defective YAP (Fig 6D).

5. Fig.4B: The authors stated that YAP and HoxA compete for binding to TEADs to inhibit YT target gene expression. They provided CHIP assay that point towards this conclusion, while the data for the competitive CoIP are not conclusive since also the amount of YAP-S127A in the input is reduced to the same extend. The question of whether shHoxA4 increases TEAD-YAP interaction needs also to be addressed. Quantifications are required and the IgG control in this particular experiment is missing.

In general input controls for immunoprecipitation experiments throughout the manuscript lack Actin or Gapdh as loading control.

We conducted CoIP assays again in HEK 293T and showed that the TEAD-YAP interaction was inhibited by HOXA4 in a dose dependent manner by quantification of protein, and a Gapdh loading control and IgG IP-control were also provided (Fig 4B). We have also shown that the TEAD-YAP interaction was increased by endogenous HOXA4 knockdown in vascular smooth muscle cells (Fig 6C). We modified Figure 4B, and added Figure 6C and an explanation on page 13, lines 3-7.

Inserted sentence (on page 13, lines 3-7):

Disruption of the TEAD-HOXA4 interaction in VSMCs by knockdown of HOXA4 significantly increased the amount of endogenous TEAD-YAP complexes (Fig 6C) and increased the occupation of HOXA4 on the TEAD-binding region in the promoters of CTGF and CYR61 but not α SMA attenuated that of phosphorylation-defective YAP (Fig 6D).

6. Fig.7: HoxA4-KO mouse and the arterial injury model

Phenotype of HoxA4-ko mice: is there overgrowth in some organs were YT-TEAD signature is increased (eg lungs) and are other Hox family members overtaking the function of HoxA4 in HoxA4-KO?

We thank the reviewer for this comment. We measured several organ weights, including the lung; however, we did not find any overgrowth of these organs (Appendix Fig S6A). Recently it was reported that deletion of TEAD-YAP transcriptional inhibitor Nerfin-1 also shows no organ overgrowth (Guo P. et.al., Elife. 2019, pii: e38843), but the reason for this has not been described. It is also difficult to show redundancy among HOX genes; however, Hoxa4/Hoxb4 double-knockout mice were embryonic lethal (Horan GS.et.al., Genes Dev. 1995; 9(13):1667-77), so Hoxb4 should play a pivotal redundant role toward Hoxa4 in the developmental stage. We now think that HOXA4 have some specific functions in conditions with highly activated YAP such as vascular injury. We added Appendix Figure S6A, and cited the work of Horan GS et al and modified the Discussion section accordingly on page19, lines 2-11.

Modified sentences (on page 19, lines 2-11):

Because repression of the YAP-TEAD interaction by HOXA4 was only observed with high YAP activity (i.e. low cell density) (Fig 4A), HOXA4 in vivo seems to be dispensable in steady-state conditions in adults. On the other hand, it may be important for the maintenance of homeostasis under YAP-activating conditions, such as cancer (Ehmer & Sage, 2016) or vascular remodeling (Wang et al, 2012). In our study, Hoxa4 KO mice show no apparent phenotype other than the previously reported mild skeletal alterations at baseline (Horan et al, 1994). A possible explanation of this finding is that other Hox genes have functional redundancy in terms of development, which is supported by a previous report of embryonic lethality observed in Hoxa4/Hoxb4 double mutant mice (Horan et al, 1995). This observation also can explain the reason why Hoxa4 KO mice show no apparent phenotype other than the previously reported mild skeletal alterations at baseline (Horan et al, 1994). In this study, we found Hoxa4 KO mice showed exacerbated neointima formation of the ligated carotid artery, in which YAP/TEAD transcriptional activity is highly upregulated (Fig 6). On the other hand, we found Hoxa4 KO mice showed exacerbated neointima formation of the ligated carotid artery, in which YAP/TEAD transcriptional activity is highly upregulated (Fig 7). This observation implies that HOXA4 has a specific molecular function and may be important for the maintenance of homeostasis in YAP-activating conditions, such as cancer (Ehmer & Sage, 2016) or

vascular remodeling (Wang et al, 2012), because repression of the YAP-TEAD interaction by HOXA4 was only observed with high YAP activity (i.e. low cell density) (Fig 4A).

7. Immunofluorescence analysis, WB as well as qRT PCRs in carotid arteries of ctr and HoxA4-KO before and after ligation (or Right-nonligated, left-ligated) should be provided to show the increase in YT-target gene expression and/or nuclear localization of YAP/TAZ before and after ligation.

We thank the reviewer for this very constructive comment. We presented immunofluorescence analysis (Fig EV5A) and WB (Fig 7F) in non-ligated and ligated carotid arteries, and confirmed the upregulation of YT-target genes as well as the downregulation of smooth muscle contractile genes in Hoxa4-deficient mice compared with WT mice, in accordance with the results of qRT-PCR. Furthermore, qRT-PCR analysis showed that differences in YT-target genes between control and Hoxa4 KO mice were also significant after ligation (Fig 7E). We added Figure 7E, 7F, EV5A and explanations on page 14, lines 17-22.

Inserted sentences (on page 14, lines 17-22):

The predominant expression of YAP/TEAD target genes in the carotid arteries of Hoxa4 KO mice was also observed at 1 week after ligation (Fig 7E). These differences in differentiation- or proliferation-associated genes in the injured artery between WT and Hoxa4 KO mice were also confirmed by immunoblotting (Fig 7F) or immunohistochemistry (Fig EV5A), leading to a two-fold increase in neointima formation in Hoxa4 KO mice at 250 μ m proximal to the ligation (Fig 7G to 7I) and at 500 μ m proximal to the ligation (Fig EV5B to 5D).

8. EVF2D: higher magnification needs to be provided as well as YAP and DAPI staining to show where HoxA4 and YAP localize in the different conditions.

We showed immunostaining of HEK 293T cells transfected with HOXA4-GFP using DAPI, anti-GFP and anti-YAP antibodies (Fig EV1E), and also GFP fluorescence with higher magnification (Fig EV1D) with different cell densities. These data suggested that HOXA4 expression did not affect YAP subcellular localization regulated by the Hippo signaling pathway. We added Figure EV1D, 1E and an explanation on page 8, line 22 – page 9, line 2.

Inserted sentence (on page 8, line 22 – page 9, line 2):

Because HOXA4 persistently stays in the nucleus without affecting YAP subcellular localization (Fig EV1D and 1E), we speculated that HOXA4 might physically interact with TEADs or YAP.

9. EVF5D: the author state that "VSMCs with nuclear HoxA4 were spindle shaped like a contractile phenotype, whereas cells without nuclear HoxA4 were rhomboid shaped like synthetic phenotype"(P12, lines 4-6). First, this is not obvious by the provided magnification it could be also dependent on the confluency. Second, which cells are those, i.e. under which conditions HoxA4 can be found in the cytoplasm and does HoxA4 have a function there?

We re-assessed HOXA4 subcellular localization in human vascular smooth muscle cells by immunostaining; however, HOXA4 existed only in the nucleus with a some reproducibility. Therefore, we modified the images (Fig EV4E) and concluded that HOXA4 constitutively stays in the nucleus. We modified Figure EV4E and deleted the sentence below.

Deleted sentence:

Of note, VSMCs with nuclear HOXA4 was spindle shaped like a contractile phenotype, whereas cells without nuclear HOXA4 was rhomboid shaped like synthetic phenotype (Fig EV5D).

Response to Referee #2

We are grateful to Referee #2 for the informative and useful comments. As described below, we have considered all of these comments and used them to improve our manuscript.

The manuscript entitled 'Homeobox A4 suppresses vascular remodeling as a novel regulator of YAP/TEAD transcriptional activity' by Kimura et al., describes the role of HoxA4 as a novel

suppressor of YAP/TEAD transcriptional activity as well as its role as a suppressor in the phenotypic switching of VSMCs. The authors have performed a thorough investigation to elucidate the molecular function of HOXA4 with proper validation. The findings are interesting for the field of YAP/TEAD and transcriptional biology. However, no evidence of the interaction of HOXA4 and TEADs leading to attenuation of YAP activity is shown in VSMCs. Furthermore, a better characterization of the HOXA4 KO mice should be provided.

We agree and thank the reviewer for the excellent and constructive comments, which have helped to considerably improve our manuscript.

-Major

1. The key mechanistic findings of the study should be shown in VSMCs as well and not only in HEK293T.

We strongly agree with the reviewer's comment, which was also pointed by the other reviewer. We also performed competitive co-IP assays and ChIP assays using human vascular smooth muscle cells and obtained consistent results with those seen in HEK 293T cells (Fig 6C and 6D). We added Figure 6C, 6D and explanations on page 13, lines 3-7.

Inserted sentence (on page 13, lines 3-7):

Disruption of the TEAD-HOXA4 interaction in VSMCs by knockdown of HOXA4 significantly increased the amount of endogenous TEAD-YAP complexes (Fig 6C) and increased the occupation of HOXA4 on the TEAD-binding region in the promoters of CTGF and CYR61 but not α SMA attenuated that of phosphorylation-defective YAP (Fig 6D).

2. Figure 2: the authors claim that the interaction between HOXA4 and TEAD would play a role during vascular remodeling; however, this interaction should be shown in VSMCs as well; and without overexpression of TEAD and HOXA4.

We strongly agree with the reviewer's comment, which was also pointed by the other reviewer. We proved that endogenous knockdown of HOXA4 increased TEAD-YAP binding in vascular smooth muscle cells using a co-IP assay (Fig 6C).

3. Figure 5G, H: counting the cell number does not really tell whether cells proliferate more or less, as other mechanisms such as cell death could be taking place. Therefore, this analysis should be replaced by a better proliferation assay (i.e. like BrdU labelling).

We agree with the reviewer on this comment. We conducted a BrdU incorporation assay in vascular smooth muscle cells with overexpression or knockdown of YAP or HOXA4, and the results were consistent with the cell count assay (Fig 5H, 5J, EV4F and EV4G). We added Figure 5H, 5J, EV4F, EV4G and an explanation on page 12, lines 13-15.

Inserted sentence (on page 12, lines 13-15):

This negative effect of HOXA4 on the proliferative capacity of VSMCs was also confirmed by bromodeoxyuridine (BrdU) incorporation analysis (Fig 5H and 5J).

4. Figure 5 (extended view): HOXA4 expression should be shown in vivo via immunostaining on tissue sections and not just based on the FANTOM5 CAGE data.

We first tried to perform Hoxa4 immunostaining on mouse tissue sections. Although we tried all commercially available antibodies against mouse Hoxa4 (described in the Methods), we found no specific antibodies against mouse Hoxa4, because several bands or non-specific staining were still observed even in Hoxa4-deficient mice. Meanwhile, we could certainly detect human HOXA4 protein, which was confirmed using both knockdown and overexpression experiments. Therefore, human aortic samples were used to confirm the expression of HOXA4 in vascular smooth muscle cells in vivo (Fig EV4B). We added Figure EV4B and an explanation on page 11, lines 10-11.

Inserted sentence (on page 11, lines 10-11):

We also confirmed the expression of HOXA4 in a human aortic tissue (Fig EV4B).

5. Figure 7: A better characterization of the HOXA4 KO mouse and in-vivo analysis needs to be shown. A staining for vSMCs and ECs would be a nice approach to show in general how vessels and SMCs look like in control and mutant embryos.

We performed immunostaining on dorsal aorta of E18.5 mouse embryos to compare *Hoxa4* KO with WT mice, which showed no difference in the staining pattern of vascular smooth muscle cells and endothelial cells (Appendix Fig S6B). We added Appendix Figure S6B and an explanation on page 13, lines 19-21.

Inserted sentence (on page 13, lines 19-21):

The contractibility of primary VSMCs harvested from *Hoxa4* KO and WT mice was also compared using a collagen gel assay and found to be almost similar (Appendix Fig S6B).

6. Could the contractibility of vSMC in control and HOXA4 mutants be measured? For example, in aortic rings?

We conducted collagen gel contraction assays to assess contractibility using primary aortic vascular smooth muscle cells harvested from WT and *Hoxa4* KO mice, which showed no difference (Appendix Fig S6C). We added Appendix Figure S6C and an explanation on page 13, line 21 to page 14, lines 1-2.

Inserted sentence (on page 13, line 21 to page 14, lines 1-2):

The contractibility of primary VSMCs harvested from *Hoxa4* KO and WT mice was also compared using a collagen gel assay and found to be almost similar (Appendix Fig S6B).

7. Figure 7B : Complementary to Panel B: In order to link comprehensively the mechanism of how HOXA4 displaces YAP, does vascular remodeling induce HOXA4 expression? which would then displace YAP in vivo, as seen in the competition assay in vitro? A staining for HOXA4 could complement the RNA expression levels after artery ligation.

We fully agree on the importance of how endogenous *Hoxa4* protein expression changes after vascular injury, and hoped to make it clear. Unfortunately, we could not find any specific anti-*Hoxa4* antibody for immunostaining on mouse tissues, although we tried all commercially available antibodies. There is only one report showing a reduction in HOXA4 comparing aortic smooth muscle cells of an aortic aneurysm with those of normal aorta by immunostaining (Lillvis JH.et.al., BMC Physiol. 2011;11:9.), implying a causal role of *Hoxa4* for the VSMC phenotype switch, because vascular smooth muscle cells are considered to transform into synthetic phenotype in aortic aneurysm (Petsophonsakul P.et.al., Arterioscler Thromb Vasc Biol. 2019 ;39(7):1351-1368., Peng H.et.al., J Am Heart Assoc. 2018;7(17):e010069.). However, this important issue should be addressed in future investigations. We added explanations in the Results and Method sections on page 13, lines 14-17, page 14, lines 14-16, and page 34, lines 12-20.

Inserted sentence (on page 13, lines 14-17):

In order to exclude the possibility that a functional truncated peptide was translated, we deleted almost all genomic regions of *Hoxa4* by inducing double-strand breaks at two sites around the start and stop codons and following homologous recombination repair (Appendix Fig S5B and 5C).

Inserted sentence (on page 14, lines 14-16):

Although we could not detect a specific signal of mouse *Hoxa4* protein due to the absence of commercially available antibodies (See Methods), the transcription of *Hoxa4* in the ligated carotid artery was significantly reduced at 1 week after operation (Fig 7C).

Inserted sentences (on page 34, lines 12-20):

Antibodies

Although all commercially-available antibodies (Abcam ab131049, Abcam ab 26097, Santa Cruz sc-398426, Santa Cruz sc-515418, MyBiosource MBS9206995, MyBiosource MBS2525887, Biorbyt orb157564) that are described as being applicable for immunoblotting and/or immunohistochemistry of mouse origin in their datasheets were tried to detect mouse *Hoxa4*, no specific signal was observed in comparisons between WT- and *Hoxa4* deficient mice-derived

samples. For the detection of human HOXA4, Abcam ab131049 was validated for immunoblotting or immunohistochemistry by knockdown and overexpression of HOXA4 in HEK 293T cells and human VSMCs.

8. Figure 7D:

- Include H&E and/or Elastica van Gieson images from the four groups: right non-ligated artery WT animals; left ligated artery WT animals; right non-ligated artery HOXA4KO; left ligated artery HOXA4KO to have a better idea on how the neointima looks in every condition with the same staining.
- Include analysis of neointima area and neointima/media ratio for the four groups mentioned before.
- Complement the morphometric analysis with the other vessel parameters: vessel area and media area between the four groups.

We agree with the reviewer on this comment. As indicated by the reviewer, we added Figure 7G to 7I, Figure EV5B to EV5D, and modified an explanation on page 14, lines 18-22.

Modified sentence (on page 13, lines 14-17):

These differences in differentiation- or proliferation-associated genes in the injured artery between WT and Hoxa4 KO mice were also confirmed by immunoblotting (Fig 7F) or immunohistochemistry (Fig EV5A), leading to a two-fold increase in neointima formation in Hoxa4 KO mice at 250 μ m proximal to the ligation (Fig 7G to 7I) and at 500 μ m proximal to the ligation (Fig EV5B to 5D).

-Minor

1. A little work on wording and structure would make manuscript sound much better.

We thank the reviewer for this advice. We have tried to improve the manuscript structure.

2. Figure 3 and figure 4B, no 'n' number is provided.

We modified the figure legends accordingly on page 47, line 4 and lines 10-11.

Inserted sentence (on page 47, line 4):

All experiments above were repeated at least twice.

Inserted sentence (on page 47, line 10-11):

B, Competitive co-IP assay and densitometry showing significantly reduced TEAD-YAP interaction by overexpression of HOXA4 in a dose-dependent manner in HEK 293T cells (n=3).

3. The authors have used student T-test in most of their analysis (comparison between 2 different variables), however, Mann whitney U test (non-parametric) was used in figure 7B. Is there a special reason for this?

There was no special reason for this; however, groups with similar variance were analyzed using parametric tests, and groups with significantly different variance were analyzed using non-parametric tests.

4. The authors mention that "HOXA4 in vivo seems to be dispensable in steady-state conditions in adults". However, this is not shown in their study. Has this been properly investigated in VSMCs in vivo in other studies? Could the authors elaborate this more in discussion and give the appropriate references?

We thank the reviewer for raising this point. The molecular function of HOXA4 in VSMCs has not been investigated previously and we have no data to conclude this; therefore, we have removed this sentence in the Discussion section from the manuscript.

Deleted sentence:

HOXA4 in vivo seems to be dispensable in steady-state conditions in adults

5. Figure 4B: It doesn't look like a dose response when expressing different levels of HOXA4. Quantification of WB should be provided.

We thank the reviewer for this comment, which was also pointed by the other reviewer. We conducted the competitive co-IP assays again and confirmed a dose-responsive decrease in TEAD-YAP binding using different levels of TEAD-HOXA4 binding with the same amount of input YAP. Quantification of immunoblotting is also shown in Fig 4B. We modified Figure 4B.

6. Figure 5E: contrast in aSMA and Calponin bands seems very high and artificial. Improve blot.

We have shown the improved images. We modified Figure 5E.

7. Typo on expanded view figure 4, panel B: should say fold CHANGE

We have corrected the manuscript accordingly. We modified Figure 4B.

8. Typo on expanded view figure 5, panel B: should say RELATIVE

We have corrected the manuscript accordingly. We modified Figure 5B.

9. Figure 2 C and D: a one-way ANOVA was performed to determine statistical differences among groups however it is not stated whether control group (YAP5SA4 -, HoxA4-) has a significant difference with nuclear YAP overexpression (YAP5SA4 +, HoxA4-), especially in the analysis of the YAP target genes CTGF and CYR61.

We have added the mark to indicate significance in the figure. We modified Figure 2C, 2D, EV1A, EV1B, EV1C, EV3B and EV3C.

10. A clearer labeling of Figure7 panels is recommended.

We have changed the labeling accordingly. We modified Figure 7A, 7B, 7D and 7E.

2nd Editorial Decision

22 January 2020

Thank you for submitting the revised version of your manuscript. It has now been seen by one of the original referees, whose comments have been pasted below. My apologies for this unusual delay in getting back to you. It took longer than anticipated to receive the referee report due to the recent holiday season.

As you can see, the referee finds that the study is significantly improved during revision and recommends publication here. Before I can accept the manuscript, I need you to address some minor points below:

-

REFEREE REPORT

Referee #2:

The authors have address most of the comments raised by this reviewer. In its current state the manuscript has been improved and it has become a comprehensive study. This reviewer has no further comments or raised issues.

2nd Revision - authors' response

31 January 2020

The authors performed all minor editorial changes.

3rd Editorial Decision

12 February 2020

Thank you for submitting your revised manuscript. I have now looked at everything and all looks fine. Therefore I am very pleased to accept your manuscript for publication in EMBO Reports.

Corresponding Author Name: Koh Ono

Manuscript Number: EMBOR-2019-48389V1